# Loss of U11/U12 spliceosome gene *ZCRB1* leads to aberrant ciliogenesis and WNT signaling

Mujeeb Ur Rehman Pirzada[1,2,*], Geralle Powell-Rodgers[1,*], Jahmiera Richee[1,*], Antto J Norppa[3], Courtney F Jungers[1], Sarah Colijn[1], Mikko J Frilander[3], Amber N Stratman[1], Sergej Djuranovic[1,2,4]

**The U12-dependent, or minor, spliceosome processes only 0.5% of human introns, and yet, it is known to profoundly influence gene expression and cellular signaling. The ZCRB1 protein is a core component of the U12 mono-snRNP, but its functional significance to minor splicing, gene regulation, and biological signaling cascades remains poorly understood. Using CRISPR-Cas9 and siRNA-targeted knockout and knockdown strategies, we show that human cell lines with a partial reduction in ZCRB1 expression exhibit significant abnormal splicing events and altered expression of minor intron–containing genes. RNA-sequencing and targeted analyses of minor intron–containing genes indicate direct mis-splicing and expression of genes involved in ciliogenesis, with a coinciding up-regulation of WNT signaling pathway components. CRISPR-Cas12a knockdown of *zcrb1* in zebrafish embryos leads to developmental patterning and body axis abnormalities, disrupted ciliogenesis, and up-regulated WNT signaling, complementing our human cell studies. This work highlights a conserved and essential biological role of the minor spliceosome, via ZCRB1, in cellular and developmental processes across species, shedding light on the molecular crosstalk that integrates splicing regulation, ciliogenesis, and WNT signaling.**

## Introduction

During mRNA maturation, splicing is a critical step that involves joining exons and excising introns, resulting in mRNA isoforms that undergo tight posttranscriptional regulation and translation into proteins (Moore et al, 1993). The spliceosome complex catalyzes splicing through two consecutive trans-esterification reactions mediated by several small nuclear ribonucleoprotein particles (snRNPs) and their associated factors (Guthrie & Patterson, 1988; Patel & Steitz, 2003; Wachtel & Manley, 2009; Hang et al, 2015). Most

metazoan species contain two parallel pre-mRNA splicing machineries called the major (U2-dependent) and the minor (U12-dependent) spliceosomes (Turunen et al, 2013). The U2-dependent spliceosome removes over 99% of human introns, categorized as major introns, whereas the minor spliceosome targets a comparatively small number of introns (~0.5%) with distinct sequence characteristics (Turunen et al, 2013). Minor introns are distinguished from the major introns by the absence of a polypyrimidine tract upstream of the 3′ splice site and the presence of a more tightly conserved 5′ splice site and branch point sequence (Larue & Roy, 2023). A major difference between the two spliceosomes is their small nuclear RNA (snRNA) composition. The major spliceosome uses the U1, U2, U4, and U6 snRNAs, whereas these components have been replaced by the U11, U12, U4atac, and U6atac snRNAs for the minor spliceosome. Only U5 snRNA is shared between the two spliceosomes (Tarn & Steitz, 1996a, 1996b; Yu & Steitz, 1997; Will et al, 2004). Although most protein components are shared between the spliceosomes, 15 proteins have been identified as specific to the minor spliceosome (Norppa et al, 2025). Among these is Zinc Finger CCHC-Type and RNA Binding Motif Containing 1 (ZCRB1; also known as U11/U12-31K), the focus of our work here.

ZCRB1 is a component of both the U12 mono-snRNP and the U11/U12 di-snRNP, which are responsible for intron recognition (Will et al, 2004; Li et al, 2024; Norppa et al, 2024). However, the mechanism by which ZCRB1 affects minor intron splicing and downstream cellular pathways remains unknown. Although minor introns comprise only a small percentage of human introns, the genes containing them are enriched in critical information-processing functions, including DNA replication and repair, transcription, translation, splicing, cytoskeletal organization, and signaling pathways (Baumgartner et al, 2019; Moyer et al, 2020). The most highly enriched functional process linked to minor intron–containing genes (MIGs) is cell-cycle and cell division regulation (Baumgartner et al, 2019). Given the necessity of these processes to sustain an organism, there has been an ongoing effort to generate minor spliceosome–specific genetic deficiency models to more deeply characterize the role of the minor spliceosome in

[1]Washington University in St. Louis, School of Medicine, Cell Biology and Physiology, St. Louis, MO, USA   [2]Brown University, Department of Molecular Biology, Cell Biology and Biochemistry, Providence, RI, USA   [3]Institute of Biotechnology, University of Helsinki, Finland   [4]University of Montenegro, Institute for Interdisciplinary and Multidisciplinary Studies, Podgorica, Montenegro

Correspondence: a.stratman@wustl.edu; sergej_djuranovic@brown.edu
*Mujeeb Ur Rehman Pirzada, Geralle Powell-Rodgers, and Jahmiera Richee contributed equally to this work

regulating eukaryotic development (Otake et al, 2002; Kim et al, 2010; Baumgartner et al, 2018).

ZCRB1 recruitment to the minor spliceosome complex is modulated by backward k-turn RNAs (bktRNAs) via methylation of the U12 snRNA. In the absence of bktRNAs, global splicing defects—including retention of minor introns—have been reported in HCT116 cells (Li et al, 2024). Similarly, a recent study on the role of DEAD-box helicase 59 (DDX59) in ciliopathy showed that depletion of ZCRB1 resulted in the retention of minor introns in HeLa cells (Che et al, 2025). In addition, functional genomic analysis of human cancer cell lines has identified the essentiality of ZCRB1 for cell survival and proliferation (Bauer et al, 2015; Wang et al, 2015; Yilmaz et al, 2018). ZCRB1 was found to have an oncogenic function in hepatocellular carcinoma by preventing the retention of intron 11 (a minor intron) of USP21, and its knockdown proved to be antitumorigenic in liver cancer (Zhang et al, 2026). However, the fundamental role of ZCRB1 in cellular homeostasis currently remains unknown.

Other minor spliceosome–specific proteins, such as RNA-binding region–containing protein 3 (RNPC3), have been more extensively studied in eukaryotic models. Loss of the zebrafish and plant orthologs of RNPC3 led to altered global gene expression and minor intron retention (Jung & Kang, 2014; Kim et al, 2010; Markmiller et al, 2014). In zebrafish, mis-splicing of Rnpc3-associated minor intron–containing genes results in significant morphological abnormalities during organogenesis, particularly affecting the development of the eyes and endodermal organs, and is lethal by 10 days postfertilization (dpf) (Markmiller et al, 2014). These findings underscore the critical importance of the minor spliceosome and associated proteins for proper development and tissue patterning events.

Loss-of-function mutations in minor spliceosome–associated proteins have significant implications for human disease. Several congenital human diseases have been explicitly linked to the dysregulation of one or more integral minor spliceosome–related snRNPs. For example, mutations in minor spliceosome U4atac snRNA cause Roifman syndrome, Taybi–Linder syndrome, Lowry–Wood syndrome, and Joubert syndrome (Norppa et al, 2025). Mutations in RNU12 cause early onset of cerebellar ataxia and CDAGS syndrome (Elsaid et al, 2017; Xing et al, 2021). In addition, mutations in RNPC3 and CENATAC components of the minor tri-snRNP complex cause isolated growth hormone deficiency and mosaic aneuploidy linked with congenital microcephaly, respectively (Argente et al, 2014; Martos-Moreno et al, 2018; De Wolf et al, 2021). Similarly, mutations in ZRSR2 and SCNM1 cause myelodysplastic and orofaciodigital syndrome (Iturrate et al, 2022; Hannes et al, 2024). Moreover, dysregulated expression of integral components of the minor spliceosome has been implicated in the etiology and progression of multiple cancers (Nishimura et al, 2022; Augspach et al, 2023). However, this list is not exhaustive, ongoing discoveries improved sequencing techniques, and more sophisticated genomic studies continue to reveal additional disorders associated with mutations in components of the minor spliceosome. Despite the critical importance of the minor spliceosome in human disease, a clear gap exists in our understanding of the consequences of U12-associated mis-splicing on signaling regulation, particularly during developmental tissue patterning events.

WNT, Sonic Hedgehog, and Notch signaling are key pathways that are essential for early development. They play crucial roles in planar cell polarity, cellular differentiation, cell proliferation, and tissue patterning (Wilson & Stoeckli, 2012). Disruption of any of these pathways can lead to embryos with laterality and anterior/posterior axis defects (Lewis et al, 2009; Hikasa & Sokol, 2013; Guzzetta et al, 2020). Interestingly, all of these pathways can be modulated through the primary cilium, an important signaling hub in nearly all cell types (Wheway et al, 2018). Primary cilia—nonmotile organelles present on the surface of most vertebrate cells—are comprised of a microtubule-based axoneme anchored by a basal body (mother centriole) and are crucial for cellular signaling and cell polarity during development. Though likely context-specific, WNT signals can be transmitted directly via the primary cilia, which act as a focal signaling platform. In addition, WNT signaling can, in some contexts, regulate ciliogenesis. This bidirectional relationship highlights the intricate and complex links between WNT activity and ciliary-mediated signaling (Kyun et al, 2020; Zhang et al, 2023; Coschiera et al, 2024).

In this study—using CRISPR-based gene knockdown models in human cells and zebrafish, combined with functional genetics, biochemistry, and transcriptomic analysis—we show that loss of ZCRB1/zcrb1 leads to altered splicing and expression of minor intron–containing genes essential for primary cilium formation and maintenance. Consequently, in ZCRB1/zcrb1-deficient human cells and zebrafish, we see a loss of primary cilia associated with an up-regulation in WNT/Wnt signaling. In zebrafish, zcrb1 loss results in body axis defects and failed gastrulation effects that can be rescued by the re-expression of WT human ZCRB1 or by treatment with a WNT inhibitor. In contrast, although loss of cilia is rescued by re-expression of human ZCRB1 in zcrb1-deficient zebrafish, WNT inhibition does not rescue ciliogenesis, indicating that the loss of cilia is not caused by Wnt dysregulation in zcrb1-deficient zebrafish. Our results reveal a novel role for the minor spliceosome gene ZCRB1, highlighting its function in splicing primary cilium–related mRNAs, its indirect control of WNT signaling, and its essential role in vertebrate development.

## Results

### ZCRB1 is required for cellular homeostasis and interacts with minor spliceosome proteins in an RNA-dependent manner

To determine the effects of ZCRB1 on cellular homeostasis and the cellular transcriptome, we performed CRISPR-Cas9 genome editing using two independent rounds of sgRNA transfection targeting exon 1 and exon 4 in HEK293 Flp-In T-REx cells to introduce a GFP donor fluorophore and genomic mutations to generate loss-of-function alleles (Fig S1A). Despite two attempts to generate ZCRB1 homozygous mutant cell lines, we only recovered clones carrying monoallelic losses (Fig S1B and C), suggesting a role for ZCRB1 in the homeostasis of HEK293 cells. We tested 16 cell lines derived from individual ZCRB1 clones for ZCRB1 protein expression. These heterozygous clones exhibited reduced ZCRB1 protein levels relative to the untreated parental HEK293 cells (Fig S2A and B). From

these, we selected four edited clones (33, 54, 63, and PL3C7) and confirmed a partial loss of ZCRB1 protein by Western blot analysis (Fig 1A) and a decrease in transcript levels by qRT-PCR, using GOLGA5 and RPL27A for normalization (selected based on their stability in our RNA-sequencing data; Figs 1B and S2C). Each CRISPR-Cas9 selected clone had a 20–70% reduction in protein levels (Fig 1A) and 30–50% reduction in steady-state ZCRB1 mRNA levels, with the greatest reduction seen in clone 54 (65%) compared with WT (Fig 1B).

To determine the impact of reduced ZCRB1 levels on global gene expression, we performed RNA sequencing on the HEK293 parental cell line versus our four biologically independent ZCRB1-heterozygous mutant clones (33, 54, 64, and PL3C7). After controlling heteroskedasticity in all samples, we observed a 2.6-fold reduction in ZCRB1 transcript levels in the ZCRB1-heterozygous mutant clones compared with the WT cells (Table S1). Differential expression analysis between the ZCRB1-heterozygous mutant clones and WT cells revealed significant dysregulation of 4,096 protein-coding RNAs (2,185 up and 1,911 down) and 389 noncoding RNAs (primarily long noncoding RNAs; 162 up and 227 down) (Fig 1C). Gene set enrichment analysis identified significant down-regulation of pathways involved in RNA metabolism and metabolic processes in ZCRB1-heterozygous mutant cells (FDR ≤ 0.05), whereas pathways related to cell signaling and tissue and organ morphogenesis were up-regulated (Fig S3, Table S1).

Among the differentially expressed transcripts were 63 spliceosome-related genes. The majority (46) were down-regulated in our RNA-sequencing dataset, including snRNP genes from both the major and minor spliceosomes (Fig 1D, Table S1). Also, several genes encoding spliceosomal snRNAs (RNUs) for the major and minor spliceosomes were identified as down-regulated by RNA sequencing. As it is often a challenge for sequencing protocols to accurately capture low-abundance, highly structured non–protein-coding RNAs, like snRNAs (Raabe et al, 2014; Boivin et al, 2020), we investigated the levels of snRNAs in ZCRB1-heterozygous clones via Northern blot analysis. These blots revealed no significant changes in major or minor spliceosome snRNA levels (Fig S4A and B), indicating that the apparent down-regulation observed by RNA sequencing likely reflects technical limitations in accurately detecting snRNAs rather than true changes in snRNA abundance. However, several protein-coding genes of spliceosome-associated proteins were identified as down-regulated by RNA sequencing for both the major and minor spliceosomes. Among the known U11/U12 di-snRNP–associated proteins, ZCRB1, SNRNP25, and SNRNP48 were down-regulated (Table S1), suggesting that ZCRB1 disruption may lead to functional protein-level changes in assembly of both the minor and major splicing machinery. Furthermore, co-immunoprecipitation revealed RNA-dependent binding of ZCRB1 with U11/U12 snRNP components, that is, RNPC3 and SNRNP48 (Fig S5).

In addition, there was a significant enrichment in down-regulated genes belonging to the snRNP core component Smith antigen (Sm) gene family and those belonging to the splicing factor 3B complex, which is responsible for branch point sequence binding in both major and minor spliceosomes (Fig 1D, Table S1) (Turunen et al, 2013). Together, these results indicate that ZCRB1 is an essential minor spliceosome factor that binds spliceosomal

components in an RNA-dependent manner, and its partial loss leads to aberrant splicing alteration pathways.

## ZCRB1 loss alters splicing of ciliary genes containing minor introns

To determine how heterozygous loss of ZCRB1 alters transcripts harboring minor introns, we analyzed the expression and mis-splicing of known minor intron–containing genes (Norppa et al, 2024). Our RNA-sequencing dataset revealed that 165 differentially regulated genes (43 up and 122 down) contained at least one minor intron (Fig 2A, Table S1). This represents 28.3% (164/579) of all known minor intron–containing genes expressed by HEK293 cells. Gene ontology analysis of these regulated minor intron harboring genes revealed a strong correlation with cilia and centrosome-related biological processes (Fig 2B). Among these minor intron–containing genes, 16 are involved in primary ciliogenesis, centrosomes, and ciliary function, including intraflagellar transport (IFT) proteins essential for cilium assembly, growth, and maintenance through protein cargo movement along the axoneme (Fig 2C, Table S1) (Scheidel & Blacque, 2018; Wang et al, 2021).

From all dysregulated genes in the RNA-sequencing dataset, we identified 116 differentially expressed cilia and centrosome-related genes, with most (77) being down-regulated (Fig 2C, Table S1). Among the IFT-B complex proteins, six were down-regulated (IFT27, IFT57, IFT172, with three being minor intron–containing genes: IFT88, IFT74, and IFT80). Notably, IFT88, a core component of the IFT-B complex, showed a 5.2-fold decrease in ZCRB1-heterozygous cells compared with WT cells (Fig 2C, Table S1) (Scheidel & Blacque, 2018; Wang et al, 2021; Lacey et al, 2023). In addition, Bardet–Biedl syndrome (BBSome) complex members TTC8, ARL6, LZTFL1, and BBS4, which are known to interact with IFT complexes for transporting signaling molecules to the ciliary membrane, were dysregulated (Fig 2C, Table S1) (Nachury, 2018). We also identified down-regulation of 10 Meckel–Gruber syndrome genes (TMEM218, RPGRIP1L, TMEM17, TCN3, TMEM67, B9D1, and minor intron–containing genes TCTN1, TMEM231, TMEM107, and TCTN2) and genes associated with nephronophthisis (NPHP3, CEP290, INVS, IQCB1) (Table S1) (Williams et al, 2011). These genes are crucial for maintaining the linkage between the basal body and the ciliary membrane, and are necessary for axoneme extension, intra-flagellar transport, and the regulation of protein movement between the cilium and cytoplasm (Fig 2C) (Sang et al, 2011; Williams et al, 2011).

Differential splicing analysis using rMATs (Wang et al, 2024) identified numerous statistically significant alternative splicing events, with skipped exons being the most common (1907 events), followed by mutually exclusive exons (644 events), retained introns (382 events), alternative 5' start sites (310 events), and alternative 3' start sites (258 events) (Fig S6A, Table S2). We identified 29 intron retention (IR) events associated with 20 unique minor intron genes (Fig S6A, Table S2), including primary cilium–associated genes such as RABL2B, ARL16, and CCDC28B (Cardenas-Rodriguez et al, 2013; Nishijima et al, 2017; Dewees et al, 2022; Wang et al, 2025). We confirmed splicing alterations in these genes by amplifying the complete coding sequence (CDS) to capture the full spectrum of events (Figs 2D and E and S6B). Although rMATs analysis did not

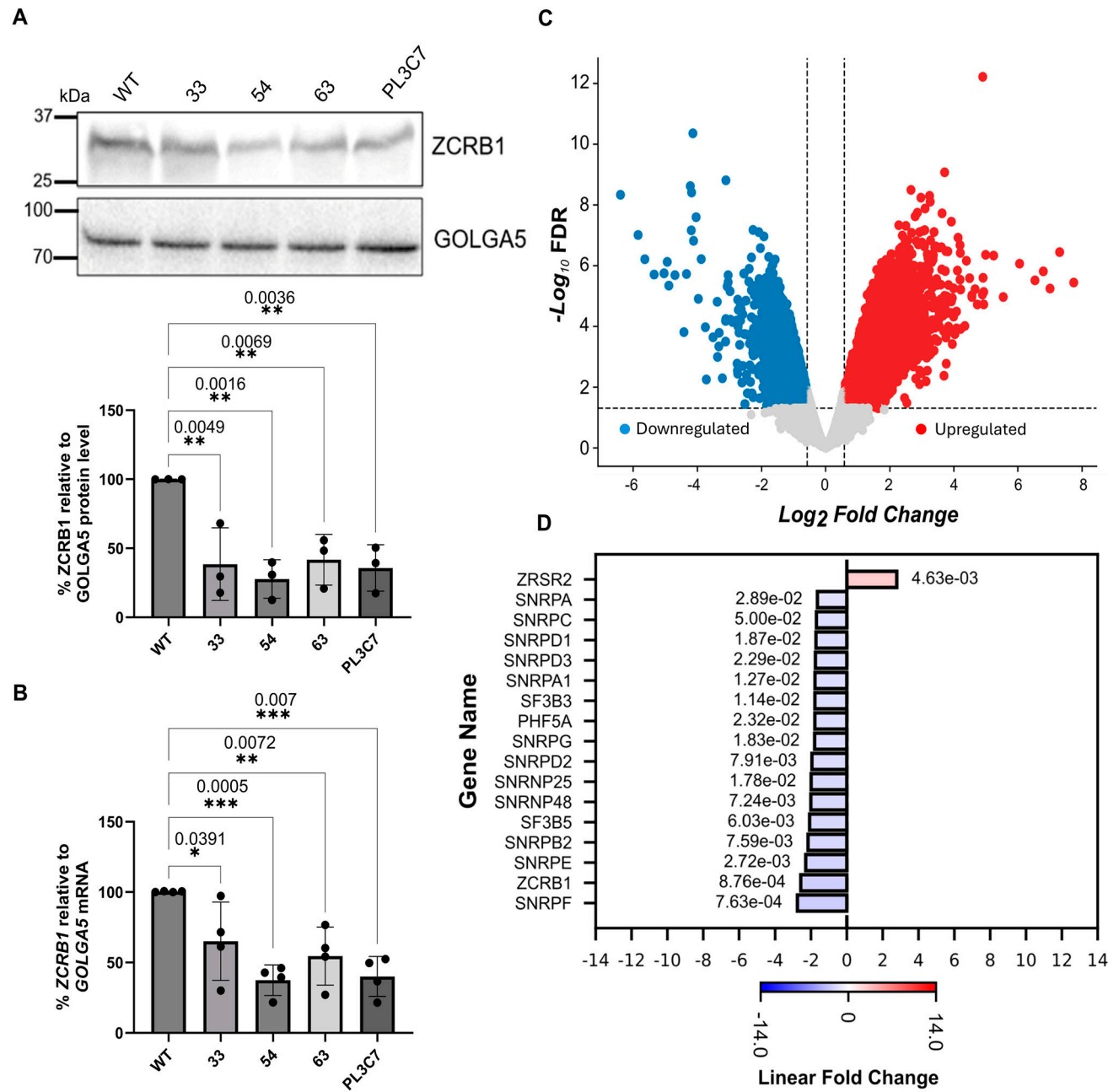

**Figure 1. ZCRB1 levels are reduced in *ZCRB1*-heterozygous HEK293 cells.**
**(A)** Western blot analysis of HEK293 Flp-In cell lines with heterozygous loss of ZCRB1. ZCRB1 protein expression is shown for parental WT HEK293 cells and the 33, 54, 63, and PL3C7 clones, quantified relative to GOLGA5 as the control. Molecular weight markers in KiloDalton (kD) are shown. **(B)** *ZCRB1* mRNA levels in heterozygous knockout clones 33, 54, 63, and PL3C7 versus WT HEK293 Flp-In cell levels as quantified by qRT-PCR. Relative mRNA levels for each clone are shown as a percentage of WT expression. Error bars in both graphs represent the mean ± s.d. *P*-values are shown and were calculated by an ordinary one-way ANOVA with Dunnett's multiple comparisons test. **(C)** Volcano plot of differentially expressed (DE) genes identified by RNA-sequencing analysis between the 4 *ZCRB1*-heterozygous HEK293 Flp-In clone cell lines and WT cells. The red and blue dots denote significantly up-regulated and down-regulated genes, respectively, whereas the gray dots represent genes with no changes in their expression (nonsignificant). The vertical dashed lines represent the cutoff for significant fold change (log$_2$ fold change of ≥ or ≤0.58), whereas the horizontal line signifies the FDR significance cutoff (≤0.05). **(D)** Bar plot of DE genes associated with both the major and minor human spliceosomes (FDR ≤ 0.05, linear FC ≥ 1.5) as identified by RNA-sequencing analysis. Red and blue bars denote up-regulated and down-regulated linear fold changes, respectively. The FDR values for each gene expression level are displayed adjacent to its respective bar.

pick up intron retention defects in the ciliary transport–associated minor intron gene *IFT88*, targeted analysis and amplification of its CDS revealed a retained intron in *ZCRB1*-heterozygous cells (Fig S6B). Across all differential splicing (DS) event types, we identified 128 events impacting transcripts associated with primary cilia and the centrosome. Of the 116 cilium-related genes showing

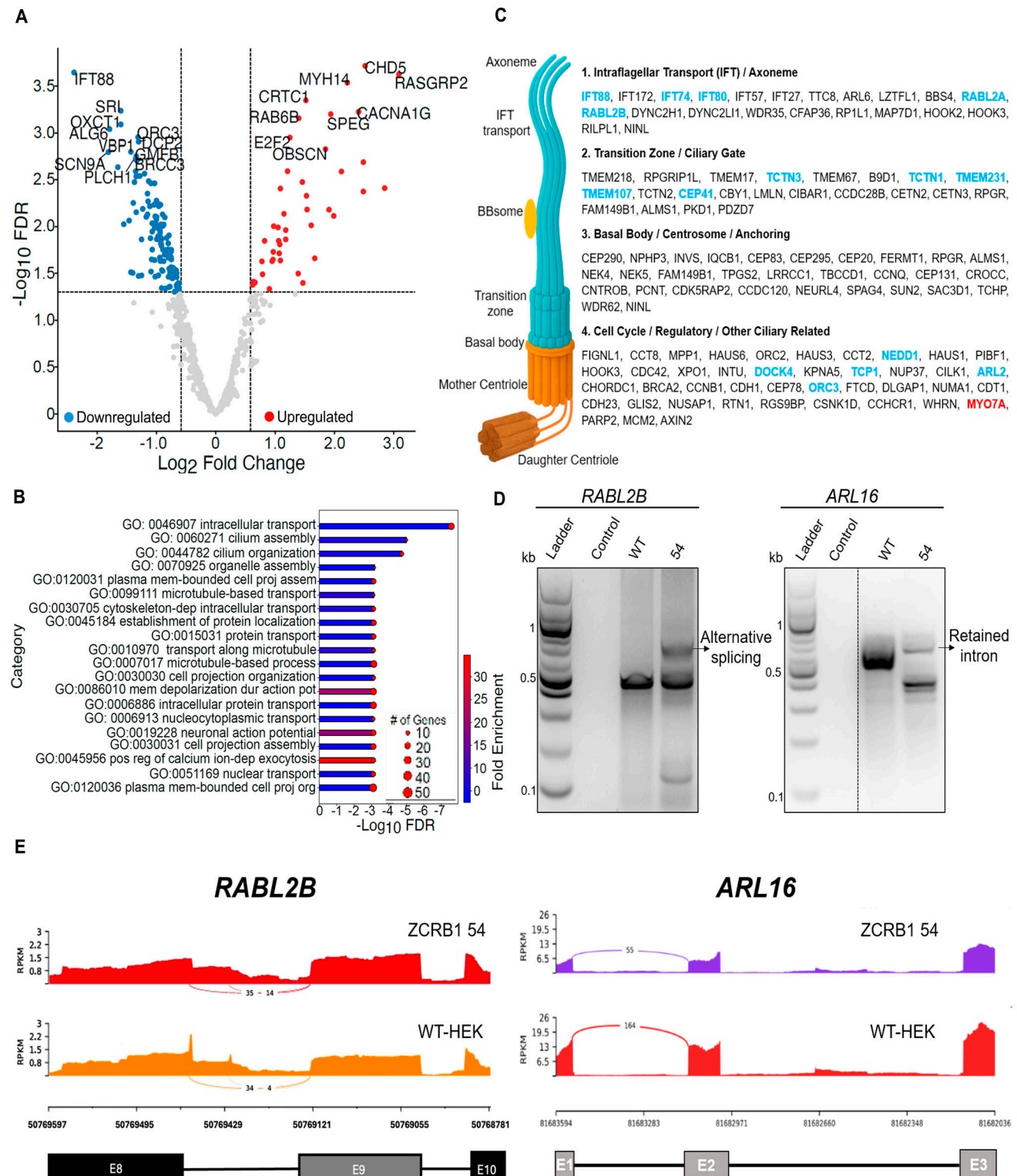

**Figure 2. ZCRB1 regulates ciliary genes by regulating the alternative splicing of minor introns.**
**(A)** Volcano plot showing DE of minor intron–containing genes as identified by RNA-sequencing analysis between the four described *ZCRB1*-heterozygous HEK293 Flp-in clones and WT cells. The red and blue dots denote significantly up-regulated and down-regulated genes, respectively, whereas the gray dots denote genes with no significant change. The vertical dashed lines represent the cutoff for significant fold change ($\log_2$ FC of ≥ 0.58 or ≤−0.58), whereas the horizontal line signifies the FDR significance cutoff (≤0.05). **(B)** Bar plot representation of the top significantly enriched biological processes identified by analysis of the DE minor intron–containing genes in *ZCRB1*-heterozygous cells versus WT cells. The length of bars is proportional to the FDR (adjusted *P*-value), and the shading within the bar represents the degree of linear fold change. The size of the red dots on the end of the bars represents the number of genes present in each identified pathway. **(C)** Schematic diagram of

differential expression (DE), 29 unique genes were both alternatively spliced and differentially expressed. Notably, more than half of these (18/29) exhibited decreased expression, suggesting that a substantial subset of cilia/centrosome genes are subjected to both transcriptional and splicing regulation under *ZCRB1*-heterozygous conditions (Table S2). Altogether, these results corroborate our omics findings and indicate that ZCRB1 contributes to the splicing of minor intron–containing genes related to ciliogenesis and ciliary trafficking.

### Loss of *ZCRB1* leads to impaired primary ciliogenesis and activation of WNT signaling

To further investigate the connection between ZCRB1 and ciliogenesis, we analyzed the effects of *ZCRB1* partial loss of function on primary cilium formation in retinal pigment epithelial-1 (RPE-1) cells, which form primary cilia more reliably than HEK293 cells (Bernatik et al, 2021). Immunofluorescence labeling of centrioles (CEP120) and the ciliary axoneme (acetylated tubulin) confirmed a decrease (66%) in primary cilium formation after *ZCRB1* siRNA treatment (Fig 3A and B). Consistent with the RNA-sequencing data, protein levels of minor intron–containing genes critical for primary cilium formation—including IFT88, CEP41, and RABL2B along with the centriolar marker CEP120—were also decreased in this RPE-1 model (Figs 3C and D and S7). These results reveal an essential role for ZCRB1 in regulating error-free splicing, particularly of procilia–related minor intron genes, validating the results of the gene set enrichment analysis of our differentially expressed minor intron–containing genes.

Gene ontology analysis of all differentially expressed genes from our *ZCRB1*-heterozygous cell RNA-sequencing analysis revealed a notable enrichment of genes in several core developmental signaling pathways (Fig S3). Detailed analysis identified differential expression of 38 genes (25 up and 13 down) associated with canonical and noncanonical WNT/Ca$^{2+}$ signaling and the planar cell polarity pathway (Table S1). Genes in the canonical WNT pathway were largely up-regulated, notably *WNT7a* (26.6-fold), *WNT9B* (5.3-fold), and *WNT11* (4.9-fold) (Fig 3E; Table S1). In addition, *AXIN1* and *AXIN2*, regulators of beta-catenin and WNT signaling, were up-regulated by 1.9 and 1.7-fold, respectively (Fig 3E, Table S1). Conversely, inhibitors of WNT signaling, such as *WIF1*, were down-regulated (10.3-fold) (Fig 3E, Table S1). Investigation of protein levels and activation of WNT pathway candidates in *ZCRB1*-heterozygous cells were inconclusive because of high variability. However, two of these mutants (63 and PL3C7) seemed to show an up-regulated trend in phosphorylated (S1490) and total LRP6 and AXIN2 protein levels (Figs 3F and S8A and B). Both of these candidates are indicative of increased WNT signaling (Wu et al, 2012; Shishido et al, 2023). These results demonstrate that the core

components necessary for the activation of canonical WNT signaling are up-regulated after *ZCRB1* loss of function.

### Zcrb1 is essential in zebrafish development and is required for cilium formation

In parallel with our investigations on the biological significance of ZCRB1 in human cells, we employed CRISPR-Cas12a genome editing in zebrafish embryos to study the role of Zcrb1 in regulating gene expression during developmental tissue patterning (Colijn et al, 2022). The human and zebrafish ZCRB1/Zcrb1 proteins share substantial sequence identity (72%) with even higher similarity in the RNA recognition motif and the zinc finger (ZF) CCHC-type domains, 76% and 94%, respectively (Fig 4A). We generated transient mosaic crispant (F0) embryos using guide RNAs (gRNAs) across a number of locations along the *zcrb1* gene (Fig S9A and B), prioritizing the use of gRNA #3, which displayed a 1.4-fold reduction in *zcrb1* transcript expression compared with Cas12a-only–injected siblings (Fig S9 and Table S3).

Fragment analysis and sequencing of the genomic sites associated with our gRNAs confirmed the generation of indels across multiple embryos, validating the efficacy of this approach (Fig S10A and B). In accordance, phenotypic analysis of 28-hour post-fertilization (hpf) embryos revealed that 55% of the *zcrb1* crispant embryos failed to gastrulate, 25% presented with disrupted dorsal–ventral body axis patterning, and 20% of the injected fish appeared phenotypically normal compared with Cas12a-only–injected siblings (Fig 4B and C). These phenotypes were rescued by reintroducing human WT *ZCRB1* mRNA into the embryos at the 1-cell stage in tandem with the *zcrb1* CRISPR gRNA, suggesting that the observed defects were specific to *zcrb1* gene editing and not off-target effects (Fig 4B and C).

Our results in human cells indicated impaired splicing and loss of primary cilium components as a primary consequence of decreased ZCRB1 function (Figs 2 and 3A–D). Therefore, to determine whether loss of primary cilium formation is a conserved phenotype in zebrafish, we generated *zcrb1* crispants on the *Tg(actb2:Mmu.Arl13b-GFP)$^{hsc5Tg}$* transgenic background, which labels all cilia in the embryo with GFP for live imaging (Borovina et al, 2010). Four conditions were assessed: Cas12a-only injected; *zcrb1* crispants; *ZCRB1* WT human mRNA–injected; and *zcrb1* crispants co-injected with *ZCRB1* WT human mRNA. Z-stack images were acquired at 32 hpf across all conditions, capturing the medial (using the dorsal aorta as a marker, magenta in the images in Fig 4D) through lateral planes of the embryo, which encompassed the skin- and skeletal muscle–associated cilia present on the embryo right side. This analysis showed a decrease in cilium numbers in the *zcrb1* crispant embryos (Fig 4D and E). Co-injection of *ZCRB1* WT human mRNA with the *zcrb1* crispant gRNA rescued cilium numbers nearly back

---

all (116) primary cilia and centrosome genes, grouped by their published function that were perturbed in *ZCRB1*-heterozygous cells. The bold ones are all minor intron–containing genes: blue ones are all down-regulated, and the red one is up-regulated. All 116 cilium-related differential proteins were manually and individually categorized based on information available in the Human Protein Atlas and other cilium-related studies. The cilium-generated image was also labeled to spatially reflect the localization of ciliary proteins. **(D)** RT–PCR results of respective complete coding sequences of *RABL2B* and *ARL16* in *ZCRB1*-heterozygous cells (clone 54). The dotted line represents a cropped lane from the same gel at the same exposure. **(E)** IGV snapshots of ciliary genes with splicing alterations indicated by rMATS. *RABL2B* shows both exon-skipping (ES) and mutually exclusive exon events, whereas *ARL16* shows mutually exclusive exon, retention of intron (RI), and ES events.

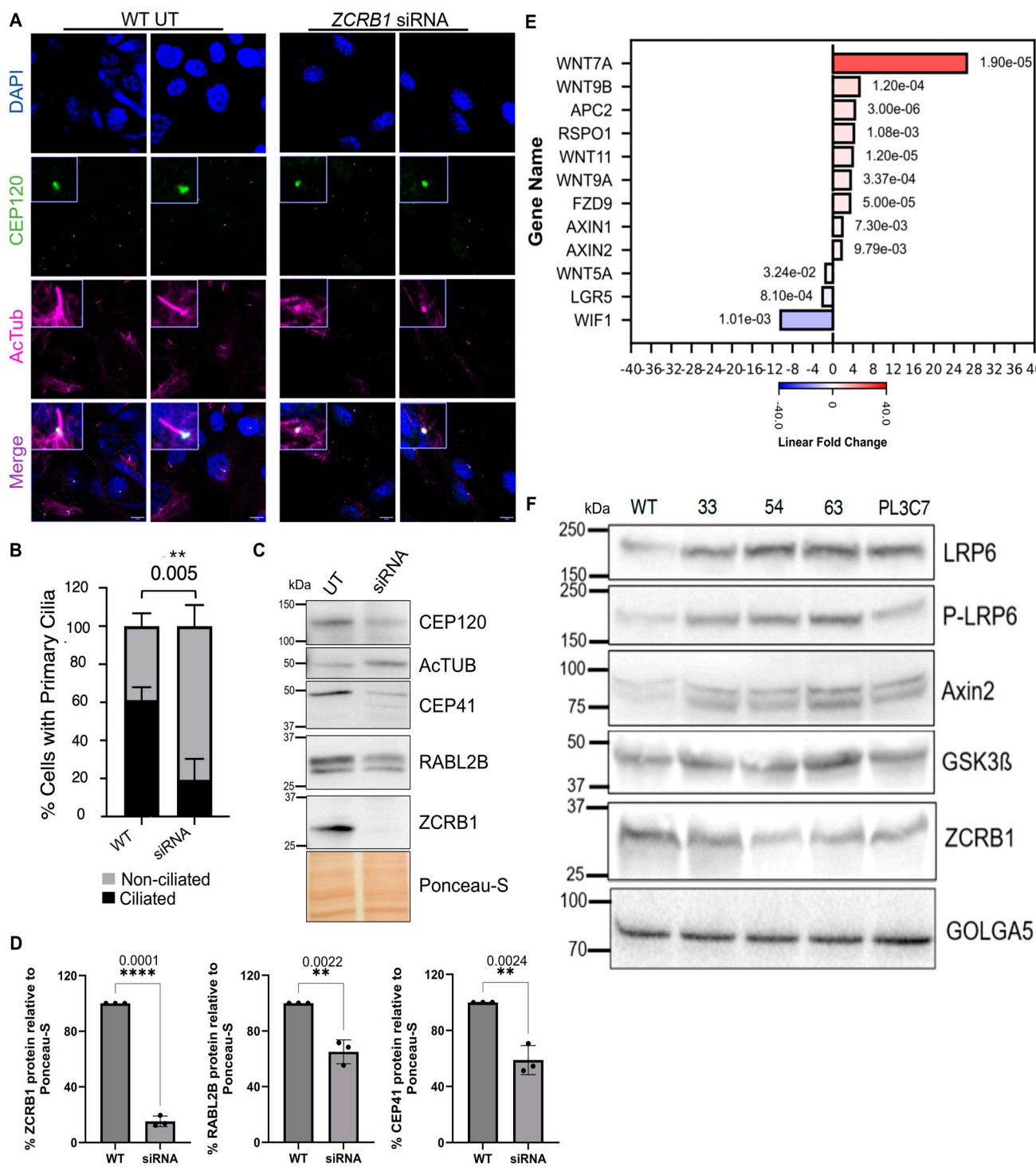

**Figure 3. Loss of ZCRB1 leads to decreased cilium formation and an up-regulation of WNT signaling.**
**(A)** Representative images from immunofluorescence labeling of centrioles (CEP-120, green) and primary cilia (acetylated tubulin, magenta) in RPE-1 cells treated with control or *ZCRB1* siRNA (scale bar: 10 μm). **(B)** Quantification of the percent number of cells (50 cells per replicate) with and without primary cilia in control versus *ZCRB1* siRNA-treated RPE-1 cells. Statistics were calculated using an unpaired *t* test. **(C)** Western blot analysis of primary cilia–associated proteins after siRNA-mediated *ZCRB1* knockdown in RPE-1 cells versus untreated (UT) control cells. **(D)** Quantification of proteins shown in C relative to Ponceau S (total protein) using a percentage scale. Error bars in all graphs represent the mean ± s.d. *P*-values are shown and measured by an unpaired *t* test. **(E)** Bar plot of canonical and noncanonical human WNT pathway genes differentially expressed by RNA sequencing between *ZCRB1*-heterozygous cells versus WT cells (FDR ≤ 0.05 and absolute linear fold change ≥1.5). Red and blue bars represent up-regulated and down-regulated linear fold changes, respectively. The color intensity within the bar denotes the degree of expression change. The FDR values for each gene are displayed next to their respective bars. **(F)** Western blot analysis of WNT signaling proteins in *ZCRB1*-heterozygous cells versus WT control cells. Vertical lanes represent samples from the WT (HEK293 Flp-In cells) and *ZCRB1*-heterozygous clones, 33, 54, 63, and PL3C7. ZCRB1 and GOLGA5 are shown

to control levels (Fig 4D and E). Across these studies, only embryos that retained a heartbeat were selected for imaging (examples are shown in Fig S11). Together, these data support the conclusion that impaired formation and/or stabilization of cilia is a conserved consequence of *zcrb1/ZCRB1* loss of function in both zebrafish and human cells.

### Impaired Zcrb1 function leads to down-regulation of cilium-related genes and increased Wnt signaling in the zebrafish

To study the effects of impaired Zcrb1 function on gene expression in zebrafish, we performed RNA-sequencing and differential gene expression analyses on crispant (F0) embryos compared with Cas12a-only–injected WT control siblings. This analysis revealed significant differential expression of 9,620 genes (6,208 up and 3,412 down) (Fig 5A, Table S3).

Gene set enrichment analysis identified up-regulated gene programs involved in cellular component biogenesis, RNA processing, and cellular and organismal development (Fig 5B, Table S3). Based on the decreased expression of ciliary genes in human cells and the loss of cilia in the zebrafish (Figs 2, 3, and 4), we performed a targeted analysis of centrosome and primary cilia–specific genes in the zebrafish RNA-sequencing data. Our analysis identified 70 differentially expressed genes related to centrosome and primary ciliary function out of the 192 genes analyzed in these pathways (Table S3). Consistent with our observations in human cells, we noted significant differential expression of several members of the canonical and noncanonical Wnt signaling pathways. Of the 122 differentially expressed zebrafish Wnt signaling pathway genes, the majority (107) were up-regulated (Table S3). In particular, Wnt pathway inhibitors, including members of the dickkopf (*dkk*) family, and *wnt5a*, which has context-specific repressor activity (van Amerongen et al, 2012), were down-regulated in *zcrb1* crispant embryos, whereas positive regulators of the Wnt signaling pathway, such as *wnt11f2* (paralog of the human *WNT11*), were up-regulated (Fig 5C, Table S3). These findings were consistent with the data seen in our human cell lines.

Given the high sequence conservation between human and zebrafish *ZCRB1/zcrb1*, along with the up-regulation of Wnt signaling after loss of *ZCRB1/zcrb1* in both models, we wanted to determine whether dysregulation of the Wnt signaling axis was underlying the gastrulation and body axis phenotypes noted in the *zcrb1* zebrafish crispants. To do so, we treated Cas12a-injected and *zcrb1* crispant embryos with the WNT inhibitor, XAV-939, and compared the phenotypes with DMSO vehicle control–treated embryos (Huang et al, 2009). The addition of XAV-939 at 20 $\mu$M to Cas12a-injected embryos led to a mild body axis phenotype, consistent with the known effects of Wnt signaling inhibition during development (Fig 6A). As noted above, the *zcrb1* crispant embryos have marked gastrulation and body axis defects (Fig 4). When treated with XAV-939 at 20 $\mu$M, the *zcrb1* crispant embryos showed a decrease in the severity of these gastrulation and axis defects and were phenotypically indistinguishable from the

Cas12a-injected, XAV-939–treated embryos (Fig 6A). In support of this finding, we used a Wnt reporter line, *Tg(7xTCF-Xla.Sia:NLS-mCherry)$^{ia5Tg}$*, to track Wnt activation in real time in zebrafish after *zcrb1* loss of function and rescue (Moro et al, 2012). Compared with Cas12a-injected control embryos, *zcrb1* crispants showed increased Wnt reporter activity, as measured by more intense mCherry fluorescence in the transgenic line (Figs 6B and C and S11). This increase in activity was rescued by co-injection of human WT *ZCRB1* mRNA into the *zcrb1* crispants, demonstrating that the altered Wnt activity is occurring in response to changes in *ZCRB1/zcrb1* levels in the embryos.

To determine whether Wnt signaling acts upstream of ciliogenesis in this context, we carried out the same experiment as outlined above, except all pharmacologic treatments were performed on the *Tg(actb2:Mmu.Arl13b-GFP)$^{hsc5Tg}$* transgenic zebrafish background to enable live imaging of cilia (Fig 6D). Although we show that treatment of *zcrb1* crispant embryos with the WNT inhibitor, XAV-939, rescued body axis phenotypes to be indistinguishable from the Cas12a-injected embryos treated with XAV-939, we observed no corresponding rescue in cilium numbers (Fig 6D and E). These results suggest that in this model, ciliogenesis defects are directly linked to the loss of *zcrb1* splicing of ciliary genes, and likely independent of Wnt signaling.

## Discussion

Despite significant strides since the identification of the U11/U12 snRNP-associated proteins over 20 years ago, their biological significance has remained understudied (Will et al, 2004; Wang et al, 2007). Here, our work provides a focused investigation into the gene regulatory function and signaling impacts of ZCRB1, expanding our knowledge of minor spliceosome–specific proteins in human biology and disease. Limited research has shown that the highly conserved ZCRB1 is an RNA-binding protein involved in SARS-CoV RNA replication and regulation of cell proliferation in glioblastoma multiforme; however, very few studies have examined its direct role in the minor spliceosome (Tan et al, 2012).

Recent research by Li et al showed that bktRNA1 (backward K-turn RNA 1) is a regulator of minor intron splicing and is necessary for the recruitment of ZCRB1 to the U12 snRNA (Li et al, 2024). A reduction in this interaction, caused by depletion of bktRNA1, led to minor intron retention. This study also demonstrated a reduction in minor intron removal efficiency through shRNA-mediated knockdown of *ZCRB1* (Li et al, 2024). In our investigation, we explored the differential splicing present between human *ZCRB1*-heterozygous mutant cells and WT HEK293 Flp-In cells using rMATs (Wang et al, 2024). Employing the same statistical framework used by Li et al, we observed dysregulation of both minor and major intron splicing (L i et al, 2024). In our pairwise comparison of cells with heterozygous expression of *ZCRB1* versus WT cells, we predictably observed a lesser degree of minor intron retention

---

again from Fig 1A as a part of a larger panel displaying WNT targets. Molecular weight markers in kD are shown. Equal protein was used for analysis, and loading was normalized to GOLGA5 levels; quantification is provided in Fig S8A and B.

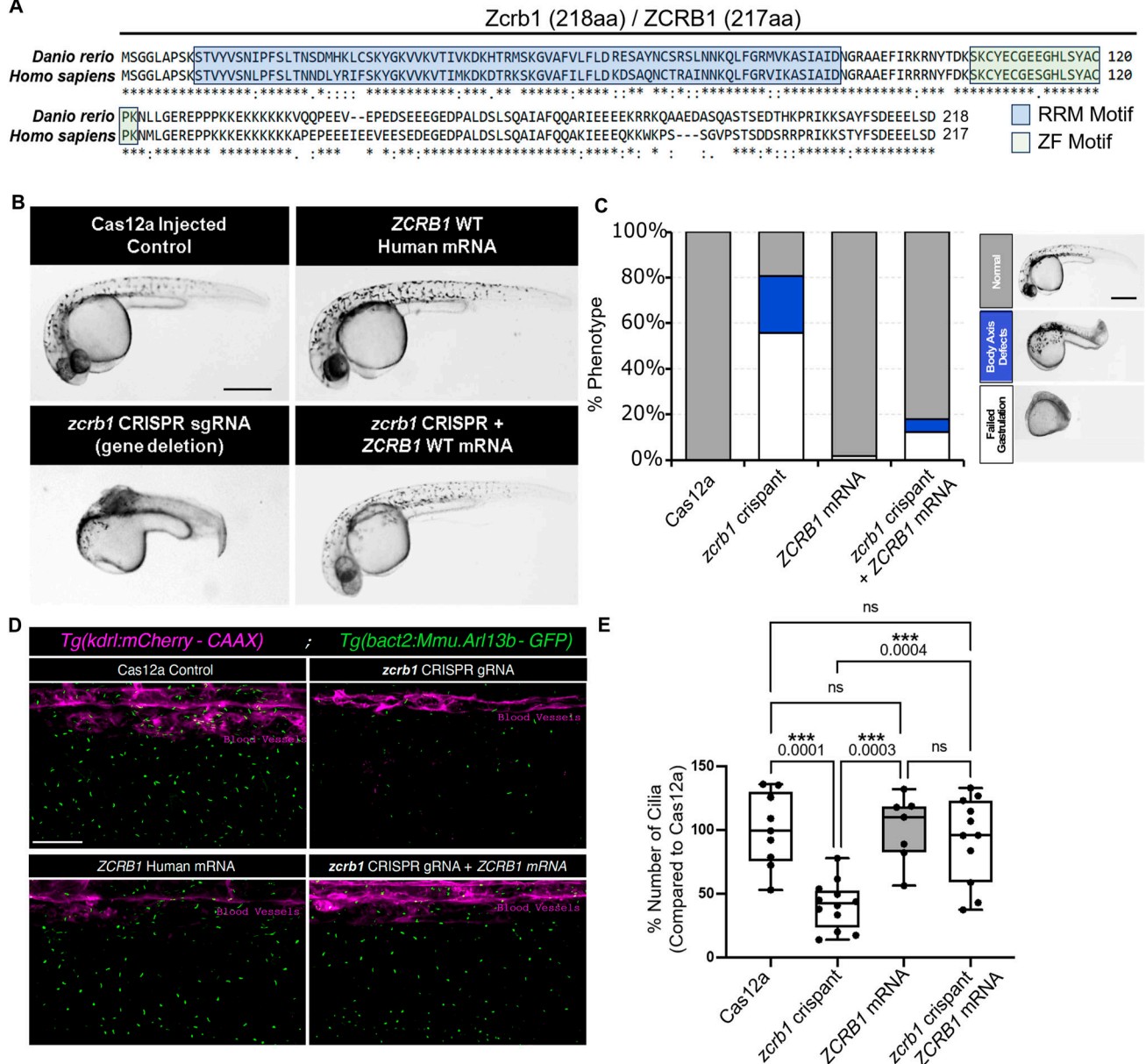

**Figure 4.  *zcrb1* is essential for development and primary ciliogenesis in zebrafish.**
**(A)** Alignment of the ZCRB1 amino acid sequence from human versus zebrafish. * represents exact homology. **(B)** Representative phenotypes associated with *zcrb1* CRISPR-Cas12a F0 crispant zebrafish embryos (CRISPR sgRNA) compared with Cas12a control–injected embryos, *ZCRB1* WT human mRNA–injected embryos, and *zcrb1* crispants plus *ZCRB1* WT human mRNA–injected embryos (rescue condition) (scale bar: 500 μm). **(C)** Stacked bar plot represents the percentage of phenotypes shown in Cas12a-injected embryos, *zcrb1* crispant–injected embryos, Cas12a embryos injected with *ZCRB1* WT human mRNA, and *zcrb1* crispant embryos co-injected with *ZCRB1* WT human mRNA. Quantification shows a phenotypic rescue of *zcrb1* crispants after re-introduction of *ZCRB1* WT human mRNA. n > 20 embryos per condition. A minimum of three independent rounds of injection were analyzed (scale bar: 500 μm). **(D)** Representative image of cilia using the *Tg(actb2:Mmu.Arl13b-GFP)[hsc5Tg]* transgenic zebrafish across Cas12a-injected embryos, *zcrb1* crispant–injected embryos, Cas12a embryos injected with *ZCRB1* WT human mRNA, and *zcrb1* crispant embryos co-injected with *ZCRB1* WT human mRNA. Cilia are shown in green; blood vessels are labeled in magenta and were used to determine consistent imaging localization (scale bar: 100 μm). **(E)** Quantification of the percent number of primary cilia between Cas12a-injected embryos, *zcrb1* crispant–injected embryos, Cas12a embryos injected with *ZCRB1* WT human mRNA, and *zcrb1* crispant embryos co-injected with *ZCRB1* WT human mRNA. All conditions are a percentage of the Cas12a control condition. Graphs are box plots with min/max error bars. *P*-values were measured by an ordinary one-way ANOVA with Tukey's multiple comparisons test. Dots represent individual embryos per condition. A minimum of two independent rounds of injection were analyzed.

than previously reported by Li et al, where their shRNA-treated cells showed more than 97% knockdown efficiency of *ZCRB1* expression (Fig S6A, Table S2) (Li et al, 2024). Our results, however, did show numerous statistically significant splicing alterations, including intron retention events, alternate 3′ and 5′ usage events, exon-skipping events, and mutually exclusive exon events

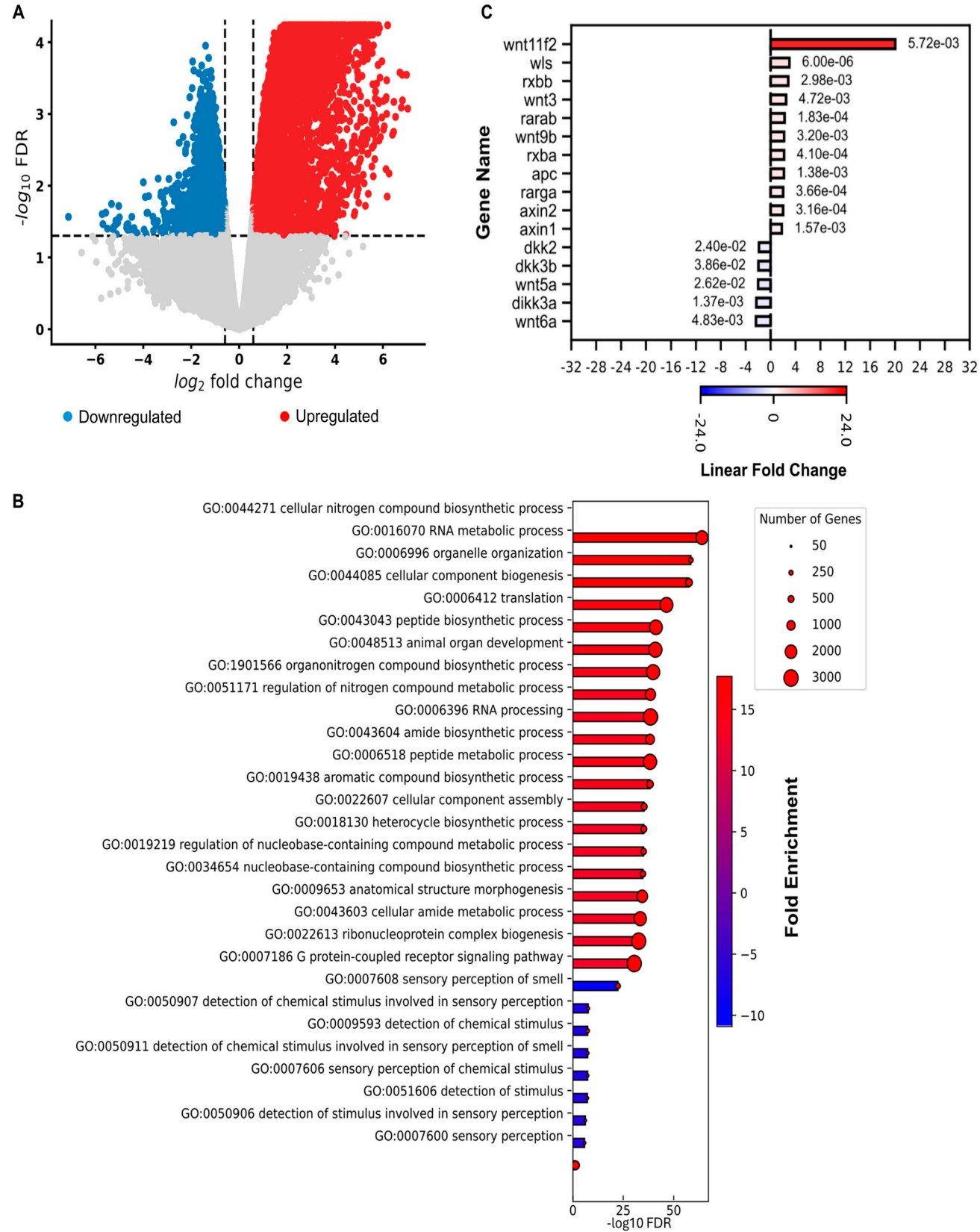

**Figure 5. Wnt signaling up-regulation after loss of *zcrb1* is conserved in zebrafish.**
**(A)** Volcano plot showing DE genes identified by RNA sequencing between 24 hpf *zcrb1* crispant embryos and Cas12a control–injected embryos. The red and blue dots denote significantly up-regulated and down-regulated genes, respectively, whereas the gray dots denote genes showing no significant change. The vertical dashed

specifically for minor intron genes and also major intron genes (Fig S6, Table S2). Interestingly, although our Northern blot analysis (Fig S4A and B) did not reveal a significant down-regulation of U12 and U1 snRNAs as seen in RNA-sequencing data (Table S1), multiple protein-coding genes associated with splice site selection and U11/U12 di-snRNP and U1 spliceosome assembly are decreased after partial loss of ZCRB1 function, suggesting that ZCRB1 levels can impact both the minor and major spliceosomes.

Beyond splicing, RNA-sequencing analyses allowed for a comprehensive analysis of the effect of heterozygous loss of *ZCRB1* broadly on gene expression. Despite roughly a 50% loss of expression in our human cell model, we observed the significant differential expression of 4,746 genes, of which 4,096 were protein-coding. Moreover, a detailed investigation of minor intron–containing genes revealed significant down-regulation in genes involved in centrosome function and essential ciliary processes, including but not limited to IFT, the formation of the BBSome, and the formation of the MKS and NPHP gene complexes. This analysis suggested that ZCRB1, as a part of the U12 spliceosome, plays a critical and targeted role in the splicing and expression of these genes and is thus necessary for ciliogenesis and ciliary function. Primary cilia are vital cellular signaling hubs, and their proper formation is crucial for maintaining coordinated cellular functions. Therefore, heterozygous loss of *ZCRB1* may have broader disease implications than previously realized, as many of the top down-regulated genes in our model, including *CEP290*, *TMEM67*, *RPGRIP1L*, and *IFT88*, are known drivers of ciliopathies (Coppieters et al, 2010; Iannicelli et al, 2010; Wiegering et al, 2018; Wang et al, 2022). The use of RPE-1 cells for the study of cilia and centrosome biology is well validated and proved to be an effective model to illustrate the significant reduction in primary cilium formation associated with ZCRB1 loss (Spalluto et al, 2013). This work allowed us to connect the decreased expression and mis-splicing of minor intron–containing ciliary genes with altered cellular structures of the primary cilia (Figs 2 and 3). Previous studies have shown that loss of *IFT88* in RPE-1 cells prevents cilium formation, supporting a possible direct role of ZCRB1 in ciliogenesis through the modulation of IFT88 expression, among other ciliogenesis genes (Simera et al, 2018). Moving forward, additional in-depth analysis of this dataset is likely to yield information regarding the direct versus indirect regulatory links between altered splicing events and gene expression, particularly via quality control pathways like the nonsense-mediated decay pathway.

Through gene ontology analysis, we pursued an unbiased approach to enhance our understanding of the functional implications of partial *ZCRB1* loss. Notably, the significantly down-regulated biological processes included ribonucleoprotein complex biogenesis and RNA processing, underscoring the role of ZCRB1 in splicing and highlighting its broader role in RNA metabolism (Fig S4, Table S1). Up-regulated biological processes included regulatory processes related to growth, differentiation, development, cell communication, and signal transduction, prompting us to explore the impact of *ZCRB1* knockdown on WNT signaling—a crucial pathway linked to proliferation, differentiation, and tissue development (Fig 3D and E, Table S1) (MacDonald et al, 2009). Although the variability between the replicates was too high to draw a firm conclusion, we observed an increased trend in phosphorylated (S1490) and total LRP6 protein, as well as AXIN2, a regulator of beta-catenin and WNT signaling, in two (63 and PL3C7) of the *ZCRB1*-heterozygous mutant cells (Figs 3E and F and S8B). Analyzing both our *ZCRB1*-heterozygous mutant cells and the zebrafish *zcrb1* crispant model, we identified a conserved up-regulation of *WNT/wnt* signaling ligands and a down-regulation of *WNT/wnt* signaling inhibitors (Figs 3E and 5C and Tables S1 and S3). Notably, increased expression of the WNT ligands *WNT11* and *WNT9* was observed in both organisms, indicating that WNT pathway overactivation is associated with heterozygous loss of *ZCRB1* (Figs 3E and F and 5C and Tables S1 and S3).

Transcriptional differences in *wnt* genes in the zebrafish manifested phenotypically as disrupted gastrulation and altered dorsal–ventral body axis polarity (Fig 4C and D and Table S3). It is established that Wnt signaling in zebrafish activates the Spemann organizer, which plays a crucial role in dorsal–ventral patterning by suppressing BMP (bone morphogenetic protein) signaling (Kiecker & Niehrs, 2001; Zou et al, 2023). This interaction helps establish a balance of Wnt signaling activity that is essential for proper embryonic patterning (Kiecker & Niehrs, 2001; Zou et al, 2023). Consistently, studies on Wnt ligand overexpression in zebrafish have shown posteriorization of embryonic tissues in proportion to increasing Wnt activity. Increased posteriorization leads to associated anterior defects—such as the absence of eyes and forebrain—a phenotype we often observed in our *zcrb1* crispants (Kiecker & Niehrs, 2001). Moreover, overactivation of Wnt signaling during embryogenesis has also been shown to negatively affect gastrulation (Kozmikova & Kozmik, 2020). Together, these observations suggest a direct link between ZCRB1-mediated transcriptional effects on *wnt* gene overexpression and impacts on zebrafish embryonic development. The ability of WT *ZCRB1* to rescue the gastrulation and body axis phenotypes seen in our model further supports a downstream functional role for altered Wnt signaling after *ZCRB1* loss.

Despite seeing an up-regulation of *WNT/wnt* signaling genes in both our human and zebrafish RNA-sequencing data—as well as functionally in the zebrafish—there remains a notable gap in the scientific literature connecting dysregulation of the minor spliceosome to alterations in *WNT/wnt* gene expression. However, this link to other developmental signaling pathways, such as Hedgehog (Hh) signaling, has been better studied. The minor spliceosome–specific factor *SCNM1*, a causative gene in the ciliopathy orofaciodigital syndrome (OFD), has been shown to

---

lines represent the cutoff for significant fold change (≥ absolute $\log_2$ fold change of 0.58), whereas the horizontal line signifies the FDR significance cutoff (≤0.05). **(B)** Bar plot representing the top enriched biological processes (BP) identified in *zcrb1* crispant embryos versus Cas12a control–injected embryos. The length of bars is proportional to the FDR (adjusted *P*-value), and the bar color represents the degree of linear fold change. The size of the red dot on the end of the bars represents the number of genes present in each identified enriched pathway. **(C)** Bar plot representation of the top DE Wnt signaling pathway genes (FDR ≤ 0.05 and linear fold change ≥ 1.5) as identified by RNA-sequencing analysis between *zcrb1* crispant embryos and Cas12a control–injected embryos. Red and blue bars represent up-regulated and down-regulated linear fold changes, respectively. The bar color intensity reflects the degree of gene expression change. The FDR values for genes are displayed next to their respective bars.

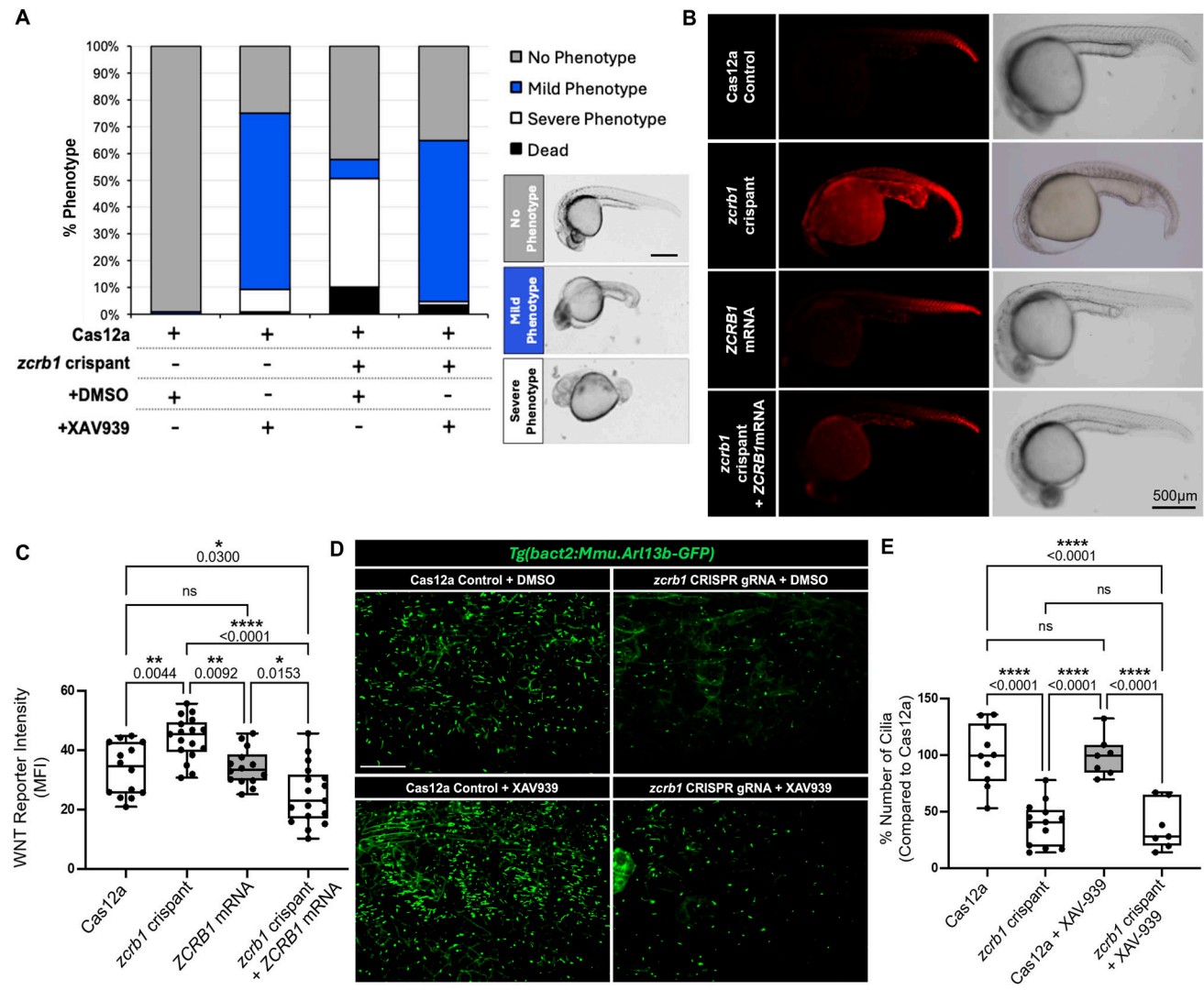

**Figure 6. Disrupted ciliogenesis caused by loss of *zcrb1* is independent of Wnt signaling up-regulation in zebrafish.**
**(A)** Quantification of gastrulation and body axis phenotypes in *zcrb1* crispant versus Cas12a control–injected embryos in the presence of the WNT inhibitor XAV-939 or DMSO vehicle control. n > 20 embryos per condition. A minimum of 3 independent rounds of injection were analyzed (scale bar: 500 μm). **(B, C)**. Representative images (B) and quantification (C) of Wnt signaling activation using the *Tg(7xTCF-Xla.Sia:NLS-mCherry)^ia5Tg* transgenic zebrafish line assessing activation in Cas12a-injected embryos, *zcrb1* crispant embryos, embryos injected with *ZCRB1* WT human mRNA, and *zcrb1* crispant embryos co-injected with *ZCRB1* WT human mRNA. The quantification represents the mean fluorescence intensity of the reporter line (arbitrary units). *P*-values were measured by an ordinary one-way ANOVA with Tukey's multiple comparisons test. Dots represent individual embryos per condition. A minimum of two independent rounds of injection were analyzed (scale bar: 500 μm). **(D, E)** Representative images (D) and quantification (E) of the percent number of primary cilia in *zcrb1* crispant versus Cas12a control–injected embryos in the presence of the WNT inhibitor XAV-939 or DMSO vehicle control. Graphs are box plots with min/max error bars. *P*-values were measured by an ordinary one-way ANOVA with Tukey's multiple comparisons test (scale bar: 100 μm).

positively regulate Hh signaling as a secondary consequence of altering cilium length (Iturrate et al, 2022). In addition, mutations to *RNU4ATAC* identified in human samples showed aberrant minor intron retention in several ciliary genes and altered ciliary function linked to activation of cAMP and Hh signaling. Accordingly, body axis and other developmental anomalies were shown in zebrafish *rnu4atac* mutants (Khatri et al, 2023). These studies link the minor spliceosome to the regulation of Hedgehog signaling, and potentially other developmental pathways, primarily by controlling the expression of ciliary genes necessary for proper ciliogenesis and ciliary function. Whether this same concept is true in the case

of WNT signaling and cilia remains an open, yet compelling question for future studies.

Through CRISPR-Cas12a targeting of *zcrb1* in transgenic GFP cilium-labeled zebrafish embryos, we demonstrated that incomplete loss of *zcrb1* is sufficient to cause the loss of primary cilium formation in zebrafish, consistent with the results from experiments in human RPE-1 cells (Figs 3A–D and 4D and E). The rescue of primary cilium formation by resupplying WT human *ZCRB1* mRNA further signifies a direct and conserved functional role for *zcrb1* in ciliogenesis (Fig 4D and E). A key finding from this work is that treatment of *zcrb1* crispant zebrafish with a WNT inhibitor, XAV-939, resulted in

the recovery of gastrulation and dorsal–ventral patterning defects (Fig 6A). This treatment, however, did not restore cilium formation, indicating that dysregulated Wnt signaling downstream of *zcrb1* loss is likely not the cause of defective ciliogenesis in the zebrafish (Fig 6D and E), implicating a more nuanced link between *ZCRB1/zcrb1*, cilia, and *WNT/wnt* gene expression. Several studies indicate that Wnt and cilia have a complex feedback mechanism in which they regulate each other, whereas others suggest that Wnt signaling is not required for ciliogenesis (Kyun et al, 2020; Bernatik et al, 2021; Zhang et al, 2023; Coschiera et al, 2024; Niehrs et al, 2024). Although our data seem to suggest that Wnt signaling is not needed for ciliogenesis in the zebrafish after *zcrb1* loss of function, we cannot rule out that the dysregulated *WNT/wnt* gene expression in the human cells and zebrafish may be a secondary consequence of cilium loss, as primary cilia can function as signaling hubs for WNT/Wnt pathways (Kyun et al, 2020; Zhang et al, 2023; Coschiera et al, 2024). Parsing these interactions will be a compelling undertaking for future studies on this topic.

Our current model suggests that heterozygous loss of *ZCRB1* in human cells and zebrafish causes the mis-splicing of ciliary genes, the up-regulation of WNT/Wnt signaling, and ultimately developmental abnormalities (Fig 7). Although this work does not resolve the controversies surrounding the relationship between WNT signaling and ciliary function, it provides valuable insights that may clarify aspects of this complex interplay while providing context for the function of the minor spliceosome in these processes. This work also provides the first detailed functional characterization of ZCRB1's role in the regulation of minor intron splicing, using the first human cell and zebrafish models of *ZCRB1/zcrb1* partial loss of function. These models offer new insights into ZCRB1 function and biological significance, while highlighting the complex mechanistic interactions between ciliogenesis, WNT signaling, and the minor spliceosome.

# Materials and Methods

### *Homo sapiens* cell culture experiments

HEK293 Flp-In T-REx and hTERT-RPE-1 cells were cultured in DMEM (Gibco) and supplemented with 10% FetalGro BGS, 5% minimum essential medium nonessential amino acids (100×, Gibco), 5% penicillin, streptomycin (Gibco), and L-glutamine (Gibco). In addition, 5 $\mu$g/ml of blasticidin and 100 $\mu$g/ml of Zeocin were added to the medium for cell line maintenance. Cells were maintained at 37°C with 5% $CO_2$. The human embryonic kidney Flp-in-T-Rex cells were obtained from Thermo Fisher Scientific (RRID: CVCL_U427, Cat. No. R78007), and the hTERT-RPE-1 cells from Moe R Mahjoub's laboratory were originally obtained from American-type culture collection (ATCC-CVCL_4388) (Bodnar et al, 1998).

### Generation of *ZCRB1*-heterozygous mutants and on- and off-target analysis of CRISPR-Cas9 editing

For CRISPR-Cas9 editing, cells were grown to 70–80% confluency. Predesigned Alt-R CRISPR-Cas9 single guide RNAs (sgRNAs)—RNA oligonucleotides containing both crRNA and tracrRNA regions—were selected for the *ZCRB1* gene from Integrated DNA Technologies (IDT). Two sgRNAs were selected based on high predicted editing performance and lower off-target risk. The Design ID, strand (+ or –), and sequence for sgRNAs 1 and 2 are written, respectively, as follows: HS.Cas9.ZCRB1.1.AA, +, GTACAAGTCATTGTTTGTCA; HS.Cas9.ZCRB1.1.AB, –, GCAAGCATTGCTATTGACAA. HEK293 Flp-In T-REx cells were co-electroporated with a 1.2-to-1 M ratio of sgRNA to CRISPR-Cas9 (GeneArt Platinum Cas9 Nuclease; Thermo Fisher Scientific) and double-stranded transgenic GFP cycle 3 donor using Neon Transfection System (Thermo Fisher Scientific). After FACS for GFP-positive cells, we expanded 80 single-cell colonies, of which 48 edited lines survived. After expansion, edited cells were assessed for ZCRB1 expression at the mRNA and protein levels, by qRT-PCR and Western blot analysis, respectively. As all of the selected clones were heterozygous for *ZCRB1*, we selected three—33, 63, and PL3C7—to cut with a second sgRNA in an independent region with the fourth coding exon (Chr12:42317406–Chr12:42317426, negative strand) (Fig S1A–C). Only one line survived from the multiple selected lines after the second CRISPR-Cas9 sgRNA complex transfection. Line 54 resulted from the second sgRNA targeting and harbors a 14-bp deletion (base 42317399) that induced a frameshift and premature stop (Fig S1A–C) on the same allele as the 1-bp deletion, leading to only monoallelic loss of *ZCRB1*.

BCFtools was used for the manipulation and parsing of CRAM files (Danecek et al, 2021). Functional characterization of single nucleotide polymorphisms was performed using SNPeff (Cingolani et al, 2012). WGS identified the creation of a *ZCRB1*-heterozygous line harboring a 1-bp insertion in the first exon of *ZCRB1* (base42324038; Chr12:42324021–4232041). This insertion led to a +1 frameshift at p.Thr22 and a premature stop codon three amino acids later (Fig S1A–C). We used Cas-OFFinder software to detect off-target events because of genomic editing by the two sgRNAs (Bae et al, 2014). Out of more than 1,000 events for each sgRNA, we found only the predicted benign intronic variant in DAB1 (Disabled-1; c.-137+177702T>C) from genomic editing with gRNA 1, supporting minimal off-target effects from the selected gRNAs (Table S4).

### Whole-genome sequencing and analysis

Libraries were constructed from 600 ng of DNA from eight CRISPR-Cas9-edited HEK293 Flp-In T-REx cells (selected at random) and controls using Kapa HyperPrep PCR-free Library Kit (Kapa Biosystems) on a SciClone NGS instrument (PerkinElmer). Genomic DNA was fragmented using a Covaris Focused-ultrasonicator with a target insert size of 350 base pairs followed by a paramagnetic bead cleanup for size selection. The library concentrations were determined using qRT-PCR (Kapa Biosystems) and pooled. Pooled libraries were sequenced to generate paired-end reads of 150 bases using the S4 300 Cycle kit with XP workflow on the Illumina NovaSeq 6000. Illumina's bcl2fastq2 software was used for base calling and demultiplexing to create sample-specific FASTQ files.

For variant calling, the samples were analyzed on DRAGEN Bio-IT processor running software version 4.0.3. Sequence reads were aligned to the GRCh38 reference genome, and alignments were generated in a CRAM format. Structural variants were called along with small and copy-number variants.

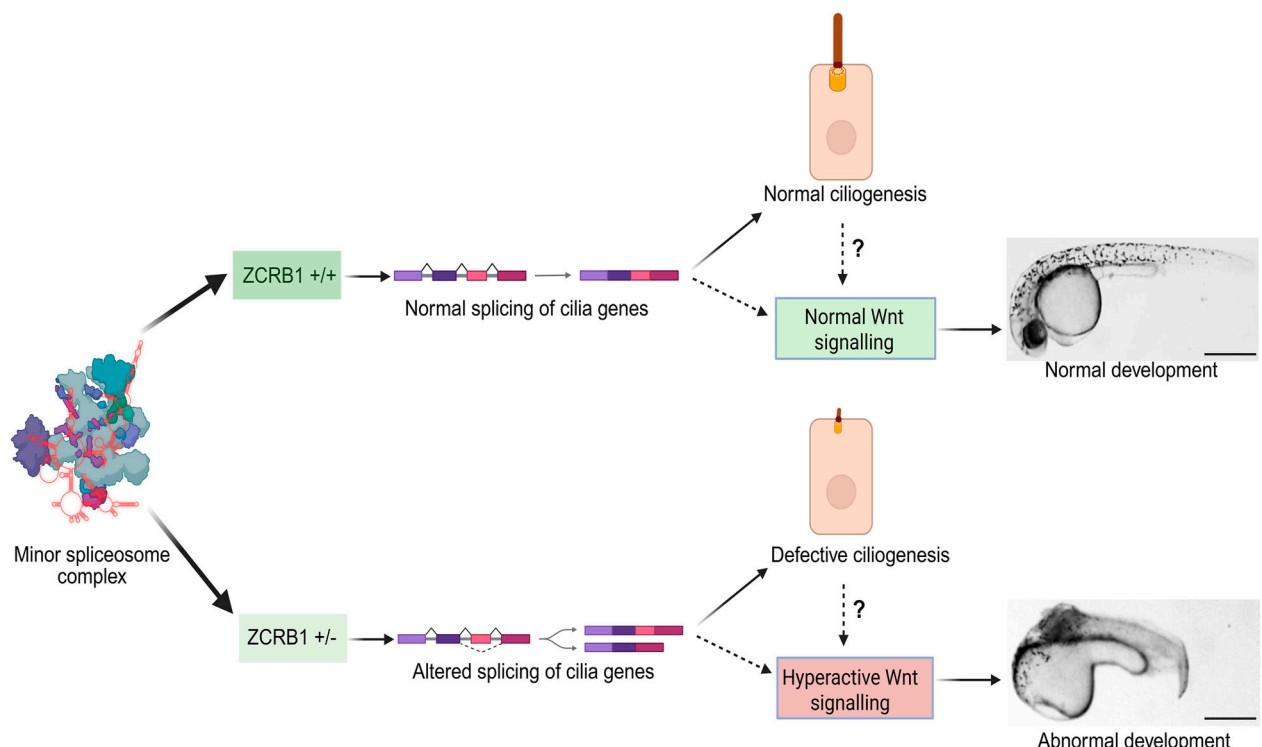

**Figure 7. Model depicting the role of ZCRB1 in ciliogenesis, WNT signaling, and vertebrate development.**
Hypomorphic levels of ZCRB1/Zcrb1 lead to altered splicing of ciliary genes with minor introns leading to loss of primary cilia and coinciding hyperactivation of WNT signaling causing developmental abnormalities. The image was generated by BioRender software.

### Protein extraction, SDS–PAGE, and immunoblotting (IB)

HEK293 Flp-In T-REx cells were lysed in an adequate amount of 1X RIPA buffer (9806, Cell Signaling (CS)) supplemented with 200 mM PMSF and 1X protease/phosphatase inhibitor (5872, CS) and passed through 1-ml syringes thrice (30 G) for efficient lysis. Supernatants were collected after performing the centrifugation at 4°C for 20 min at 20,000 G. Protein estimation was performed by the Bradford assay (5000006; Bio-Rad), and an adequate amount was loaded on SDS–PAGE gradient gels (3450123; BR). The samples were dissolved in sample loading buffer (1610791; BR) supplemented with reducing agent (1610792; BR) and subjected to run at constant 150V in the presence of XT MES running buffer (1610789; BR). Separated proteins were blotted on a polyvinylidene fluoride (PVDF) membrane by semi-dry transfer at 25V for 30 min. The membranes were blocked with 5% milk in PBS-T (0.1% Triton in PBS) for 30 min and incubated with primary antibodies either for 2 hr at RT or overnight in the cold room at constant shaking. Secondary HRP-conjugated antibodies were incubated for 1 hr at RT, then the antibody incubation membranes were washed thrice with PBS-T for 5 min, and protein levels were detected by chemiluminescence (34577; BR). The following primary antibodies were used in this study: ZCRB1 (25629-1-AP; Proteintech), LRP6 (2560, CS), phospho-LRP6 (Ser1490) (2568, CS), IFT88 (13967-1-AP, Proteintech), GSK3ß (9315; CS), ß-actin (4967; CS), RABL2B (11588-1-AP; Proteintech), acetylated tubulin (T6793; Sigma-Aldrich), and CEP41 (PA5-103727; Invitrogen). CEP120 Ab was synthesized in Moe

R. Mahjoub's laboratory at WashU (PMID: 35443171) and was kindly shared by him. Secondary antibodies include HRP-conjugated anti-rabbit IgG (7074; CS) and anti-mouse IgG (7076; CS) for Western blotting. For immunofluorescence, Invitrogen anti-rabbit (A11008) and anti-mouse (A11005) secondary antibodies were used in this study.

### RNA isolation, cDNA synthesis, and quantitative and real-time PCR

RNA was extracted using the RNeasy mini kit (74104; QIAGEN) by following the manufacturer's instructions. The extracted RNA was treated with TURBO DNase (AM1907; Thermo Fisher Scientific) followed by cDNA synthesis using iScript Advanced cDNA Synthesis Kit (1725037; BR). The qRT-PCR was performed using iQ SYBR Green Supermix (1708880; BR) by following the manufacturer's instructions in a c1000 thermocycler (CFX96; BR). The qRT-PCR data were normalized to housekeeping genes, namely, *GOLGA5* and *RPL27A*, based on our RNA-sequencing dataset. Three technical replicates per reaction and at least three biological replicates per experiment were performed. The following primers were used for qRT-PCR amplification and were designed by the Primer3 webtool. The length of primers was kept between 18 and 23 nt and the Tm between 57 and 62°C with GC% between 30 and 70:

*ZCRB1* (NM_033114) forward primer (FP): CTCCAAGTAAGAGCACAG TGT.

*ZCRB1* reverse primer (RP): CAACCCCTTTACTCTTCCTGG.

*GOLGA5* (NM_005113.4) FP: CGAACAGCAGATGAACTCCG.
*GOLGA5* RP: AGCTTTGCGAACTTTTCCGT.
*RPL27A* (NM_000990.5) FP: CTTCTGCCCAACTGTCAACC.
*RPL27A* RP TTTGGCCTTCACGATGACAG.

The following pairs of primers were used to amplify the CDS of genes for validating splicing alterations in minor intron genes:

*CCDC28B* (NM_024296.5) FP: ATGGATGACAAAAAGAAGAAACGGAGT CCCAAG.
*CCDC28B* RP: CTACGCAGCGGACTGCTCC.
*IFT88* (NM_175605) FP: ATGAAATTCACAAACACTAAGGTAC.
*IFT88* RP: TTATTCTGGAAGCAAATCATCTC.
*RABL2B* (NM_001003789) FP: ATGG CAGAAGACAAAACCAAACCG.
*RABL2B* RP: TCAGCTGTGGGGAGAGGCC.
*ARL16* (NM_001040025.3) FP: ATGTGTCTCCTGCTGGGGG.
*ARL16* RP: TCAATCGTTGGCTCTGTGGGTGG.

## Small interfering RNA (siRNA) transfections, immunofluorescence, and microscopy

Transient depletion of *ZCRB1* in RPE-1 cells was performed by electroporating an equimolar cocktail of siRNAs 2 and 3 (150 PM). The siRNA treatment was performed for 100 hr by electroporating the cells twice at 0 hr and at 50 hr. The electroporation was performed by the Neon electroporation system (MPK5000; Invitrogen). The *ZCRB1* siRNAs were obtained from IDT targeting the CDS region of *ZCRB1* mRNA with the following sequence, siRNA2: 5′-GCAAGCAUUGCUAUUGACAAUGGAA-3′, and siRNA3: 5′-CCCUCAA-CAUCAGAUGAUUCAAGAC-3′. To induce the formation of primary cilia, the cells were serum-starved (0.5%) for 48 h before performing the IF and IB (Takahashi et al, 2018). TYE 563 transfection control (not shown), HPRT-s1 positive control, and negative control siRNAs from the provider were used to validate the efficiency and specificity of the siRNA setup. For IF, the cells were seeded to achieve 80% confluency before fixation on a glass coverslip (12541007; Thermo Fisher Scientific). After ice-cold methanol fixation for 20 min at −20°C, cells were rinsed thrice with 1X PBS and permeabilized with 0.1% Triton X-100 in PBS (PBS-T) for 20 min at RT, followed by blocking with 3% BSA (A2153; Sigma-Aldrich) in PBS-T at RT. The samples were incubated with CEP120 (generated in Moe R Mahjoub's laboratory) and acetylated tubulin (T6793; Sigma-Aldrich) primary antibodies overnight in PBS-T. After three PBS-T washes, the samples were incubated with Alexa Fluor dye–conjugated respective secondary antibodies for 1 hr at RT. The nuclei were stained with DAPI (62248; Thermo Fisher Scientific), and the samples were mounted with cytoseal XYL (8312-4, Epredia). The excitation/emission spectrum for DAPI, Alexa Fluor 488, and Alexa Fluor 594 (shown in magenta) is 358/461 nm, 495/519 nm, and 590/617 nm, respectively (Fig 3A). Images were captured with a Zeiss LSM 880 confocal microscope using 100X oil objective and processed with Fiji/ImageJ software.

## Zebrafish work (animal work), CRISPR-Cas12a, and mRNA injections into zebrafish embryos

Zebrafish husbandry and research protocols were reviewed and approved by the Washington University Animal Care and Use Committee (#24-0434). All animal studies were performed according to Washington University–approved protocols, in compliance with the Guide for the Care and Use of Laboratory Animals. Guide RNAs (gRNAs) to zebrafish *zcrb1* were identified using CHOPCHOP (Montague et al, 2014; Labun et al, 2016, 2019).

sgRNA #1 (Cas9): 5′-GCAGCCTGAAGAAGTGTAAG-3′
sgRNA #2: 5′-TTCCAGGGATAAAGATGAGTGGTGGTTT-3′
sgRNA #3: 5′-TTCCGACAGTTATGCTCAAAATATGGCA-3′
sgRNA #4: 5′-TTCTCGCTCATGAACAACAAACAGGTA-3′

10 nmol of Alt-R A.s. Cas12a crRNA was ordered from IDT (as a presynthesized gRNA) for injection with 100 ng/pl Cas12a (Cpf1) for editing (IDT Alt-R A.s. Cas12a. [Cpf1] Ultra, #10001272). CRISPR-Cas12a mixtures were injected into wild-type AB* zebrafish embryos at the 1-cell stage (Labun et al, 2019; Colijn et al, 2022).

Human *ZCRB1* mRNA was generated by in vitro transcription of pcDNA-DEST40 reporter constructs containing the coding transcript of *ZCRB1* using mMESSAGE mMACHINE T7 Transcription Kit. mRNA was treated with TURBO DNase (AM1907; Thermo Fisher Scientific) and purified by ethanol precipitation with 3M sodium acetate before injection into wild-type AB* or co-injected with gRNA into wild-type AB* zebrafish embryos at the 1-cell stage (Green & Sambrook, 2020). Mixtures to be injected as 1-nl boluses into zebrafish 1-cell stage embryos included the following:

*zcrb1* gRNA: 0.85 µl of 10 nmol individual stock gRNA, 1.0 µl of 100 µg stock Cas12a (Cpf1), 1.95 µl 2M KCl, 5.0 µl H2O, 1.0 µl phenol red.

*zcrb1* gRNA + WT *ZCRB1* mRNA: 0.85 µl of 10 nmol individual stock gRNA, 1.0 µl of 100 µg stock Cas12a (Cpf1), 1.95 µl 2M KCl, 0.25 µl H2O, 5.0 µl WT ZCRB1 mRNA, 1.0 µl phenol red.

Cas12a (Cpf1)-only control: 1.0 µl of 100 µg stock Cas12a (Cpf1), 1.95 µl 2M KCl, 6.05 µl H2O, 1.0 µl phenol red.

WT *ZCRB1* mRNA only: 1.95 µl 2M KCl, 2.05 µl H2O, 5.0 µl WT ZCRB1 mRNA, 1.0 µl phenol red.

WT *ZCRB1* mRNA and phenol red were added after a 10-min incubation of the gRNA/Cas12a mixture at 37°C. The mixture was then kept on ice until microinjection into the zebrafish embryos at the one-cell stage.

## Northern blotting of spliceosome components

TRIzol-extracted total RNA (3 µg) was resolved on a 7% urea–polyacrylamide gel, transferred overnight at 150 mA onto Zeta-Probe (Bio-Rad) membrane using an Owl VEP-2 blotter, and cross-linked using 254-nm UV light. Oligonucleotide probes (Table S5) were 5′ end–labeled with [γ-$^{32}$P]-ATP using T4 Polynucleotide Kinase (NEB). Hybridization was carried out overnight at 42°C (DNA oligos) or 45°C (LNA/DNA oligos) in buffer containing 6xSSC, 25 mM Na$_2$HPO$_4$/NaH$_2$PO$_4$ (pH 7.4), 0.5% SDS, 5x Denhardt's solution, 150 µg/ml yeast RNA (Roche), 5 × 10$^6$ cpm of probe, and 50% formamide (for LNA/DNA probes only). The membrane was washed sequentially with 2× SSC, 0.1% SDS, and 0.5× SSC, 0.1% SDS at room temperature for 15 min each. For LNA/DNA probes, additional washes were carried out with 0.1× SSC, 0.1% SDS at room temperature and 60°C. The blot was exposed on an imaging plate and scanned using Sapphire FL Biomolecular Imager. For reprobing, the membrane was stripped with 0.01×SSC, 0.5% SDS at 95°C for 2× 15 min.

## Imaging of cilia and *wnt* transgenic zebrafish reporters

Transgenic zebrafish lines used in this work were *Tg(actb2:Mmu.Arl13b-GFP)$^{hsc5Tg}$* and *Tg(7xTCF-Xla.Sia:NLS-mCherry)$^{ia5Tg}$* (Borovina et al, 2010; Moro et al, 2012). *Tg(actb2:Mmu.Arl13b-GFP)$^{hsc5Tg}$* or *Tg(7xTCF-Xla.Sia:NLS-mCherry)$^{ia5Tg}$* adult zebrafish were crossed and injected with the gRNA/Cas12a mixtures as described above. After injection, the embryos were housed at 28°C for 28 h for the assessment of phenotypes. In all cases, the embryos were allowed to mature for ~3 h postinjection and unfertilized embryos removed to aid in accurately assessing gastrulation and live/death ratios after injections. Embryos were grown for ~28 h before phenotypic and genotypic analysis.

Imaging was done using a Ti2 Nikon microscope with CSU-W1 confocal scanner, 20× or 40× APO-Plan objective, and a Fusion camera to image cilia and Wnt activation within zebrafish embryos. A Z-stack step size of 0.4 $\mu$M was used for all acquisitions. At 24 hpf, embryos were dechorionated using Dumont Tweezers, Style 55 (#72707-01; Electron Microscopy Sciences). The embryos were immobilized using MS-222 and embedded in 1% UltraPure Low Melting Point Agarose (#16520050; Thermo Fisher Scientific) and fish water for live imaging. Only embryos that retained a heartbeat were used for imaging analysis and quantification of Wnt reporter and cilium phenotypes.

## Genotyping and fragment analysis of *zcrb1* gRNA–injected embryos

Embryos were placed individually in PCR tubes, and genomic DNA (gDNA) was extracted using equal volumes of extraction solution (#SLCQ0691; Sigma-Aldrich) and tissue preparation solution (#SLCH8297; Sigma-Aldrich). Samples were incubated in SimpliAmp Thermal Cycler (Thermo Fisher Scientific) for 15 min at 55°C, followed by 15 min at 95°C to allow for tissue digestion. Primers flanking the *zcrb1* gRNA sites were used to generate small amplicons for analysis of CRISPR cutting efficiency using Agilent 5300 Fragment Analyzer (Colijn et al, 2022). Primers used were as follows:

gRNA #2: FP-AAGGATCATGTTTTGCATGTTG, RP-ATGATCATC-CACTTGGCAGTTA. gRNA #3: FP-GGCCTAAAAATAACAAAGCATCA, RP-TCAAGCCATGCATTGACATAAA.

gRNA #4: FP-GATGAGTAAAGGAGTGGCGTTC, RP-CACACCTGAACCAGCTAATCAA.

## RNA sequencing and analysis

RNA was extracted using the RNeasy system (QIAGEN) for human cell lines and Zebrafish. RNA concentration was measured by NanoDrop (Thermo Fisher Scientific). Total RNA integrity was determined using Agilent Bioanalyzer or 4200 TapeStation. Library preparation was performed with 1ug of total RNA. Ribosomal RNA was removed by an RNase-H method using RiboErase kits (Kapa Biosystems) for human cell lines and by a hybridization method using Ribo-ZERO kits (Illumina-EpiCentre) for zebrafish samples. Among this pool of depleted rRNA, mRNA (without explicit polyA+ purification) was then fragmented in reverse transcriptase buffer and by heating to 94 degrees for 8 min. mRNA was reverse-transcribed to yield cDNA using the SuperScript III RT enzyme (Life Technologies, per the manufacturer's instructions) and random hexamers. A second strand reaction was performed to yield ds-cDNA. cDNA was blunt-ended, had an A base added to the 3' ends, and then had Illumina sequencing adapters ligated to the ends. Ligated fragments were then amplified for 12–15 cycles using primers incorporating unique dual index tags. Fragments were sequenced on an Illumina NovaSeq 6000 using paired-end reads extending 150 bases. Basecalls and demultiplexing were performed with Illumina's bcl2fastq software with a maximum of one mismatch in the indexing read. RNA-sequencing reads were then aligned to the Ensembl release 101 primary assembly with STAR version 2.7.9a1. Gene counts were derived from the number of uniquely aligned unambiguous reads by Subread:featureCount version 2.0.32. Isoform expression of known Ensembl transcripts was quantified with Salmon version 1.5.23. The ribosomal fraction, known junction saturation, and read distribution over known gene models were quantified with RSeQC version 4.04. TMM normalization size factors were calculated using the R/Bioconductor package edgeR (Robinson et al, 2010). Ribosomal genes and genes not expressed in the smallest group size minus one samples greater than one count-per-million were excluded from further analysis. The TMM size factors and the matrix of counts were then imported into the R/Bioconductor package Limma (Ritchie et al, 2015). Differential expression analysis was performed using Limma to analyze for differences between controls and CRISPR-edited human and zebrafish samples, respectively. Gene set enrichment was performed using GAGE (Luo et al, 2009). Statistically significant genes were considered to have an adjusted *P*-value (FDR) < 0.05 and a log2FC <−0.58 or >0.58. GO terms were filtered for pathways scored with adjusted *P*-values (FDR) < 0.05.

## Differential splicing analysis

Alternative splicing analysis for human samples was performed using the rMATS (turbo0.1) on a cloud computing platform. BAM files generated from sequencing alignments as previously described were used for analysis. Alternative splicing events, including exon skipping (SE), mutually exclusive exons (MXE), retained introns (RI), alternative 5' splice sites (A5SS), and alternative 3' splice sites (A3SS), were identified for each sample and then compared across conditions to detect differences in alternative splicing events. Alternative splicing events were filtered for events containing adjusted *P*-value (FDR) < 0.05 and delta PSI > 0.05. Gene annotation for major and minor human intron–containing genes was performed using a list of human introns as described by Norppa et al. Functional enrichment of MIGs was performed using ShinyGO v0.80 ("ShinyGO: a graphical gene-set enrichment tool for animals and plants | Bioinformatics | Oxford Academic," n.d.). GO terms were filtered for pathways scored with adjusted *P*-values (FDR) < 0.05 (Table S2).

## Statistical analysis

Statistical analyses for Figs 1A and B, 3B and D, 4E, 6C and E, S2C, and S7 were performed using GraphPad Prism 11 with more details in corresponding figure legends. Data normality was determined

using the Shapiro–Wilk normality test. Statistical analyses, post hoc tests, and *P*-values are all described in corresponding figures and figure legends. Significance was determined by a *P*-value of 0.05 or less. The FDR and *P*-values associated with the RNA-sequencing data were calculated with Limma software. Fisher's exact testing was performed using the scipy.stats module from the SciPy library (Virtanen et al, 2020).

## Data Availability

All data are available in the main text or the supplementary materials. The RNA-sequencing and WGS data can be found at SRA database ID PRJNA1442218.

## Supplementary Information

## Acknowledgements

We thank the Genome Technology Access Center at McDonnell Genome Institute, Washington University School of Medicine, for performing the RNA and whole-genome sequencing. We thank the Center for Cellular Imaging, Washington University, for providing us imaging microscopes. We thank Moe R. Mahjoub for kindly sharing the RPE-1 cells with us. The work in the Djuranovic laboratory is supported by NIGMS R01GM136823 and R01GM112824, the Chen-Zuckerberg Neurodegeneration Initiative, and the Siteman Cancer Center Investment Award. Work in the Stratman laboratory is supported by NIGMS R35GM137976 (AN Stratman) and NHLBI K99HL171944 (S Colijn). Work in the Frilander laboratory was supported by grants from Academy of Finland (341477) and Sigrid Jusélius Foundation.

### Author Contributions

MUR Pirzada: conceptualization, data curation, formal analysis, validation, investigation, methodology, and writing—original draft, review, and editing.
G Powell-Rodgers: conceptualization, data curation, formal analysis, investigation, methodology, and writing—original draft, review, and editing.
J Richee: validation, visualization, and methodology.
AJ Norpa: methodology.
CF Jungers: investigation.
S Colijn: data curation, formal analysis, and writing—original draft, review, and editing.
MJ Frilander: conceptualization, investigation, and writing—review and editing.
AN Stratman: conceptualization, supervision, funding acquisition, validation, visualization, methodology, project administration, and writing—original draft, review, and editing.
S Djuranovic: conceptualization, formal analysis, supervision, funding acquisition, validation, investigation, visualization, methodology, project administration, and writing—original draft, review, and editing.

### Conflict of Interest Statement

The authors declare that they have no conflict of interest.

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
