## [Reviewer comments · Life Science Alliance]

Loss of U11/U12 spliceosome gene ZCRB1 leads to aberrant Ciliogenesis and WNT signalling

Mujeeb Ur Rehman Pirzada, Geralde Powell-Rodgers, Jahmiera Richee, Antto Norpa, Courtney Jungers, Sarah Colijn, Mikko Frilander, Amber Stratman, and Sergej Djuranovic

DOI: <https://doi.org/10.26508/lsa.202503607>

Corresponding author(s): Sergej Djuranovic, Brown University and Amber Stratman, Washington University in St. Louis

Review Timeline:

Submission Date:	2025-12-18
Editorial Decision:	2026-02-11
Revision Received:	2026-03-27
Editorial Decision:	2026-04-28
Revision Received:	2026-05-11
Accepted:	2026-05-15

Scientific Editor: Sarita Hebbar

Transaction Report:

February 11, 2026

Re: Life Science Alliance manuscript #LSA-2025-03607-T

Sergej Djuranovic
Brown University
Molecular Biology, Cell Biology and Biochemistry
Providence, RHODE ISLAND

Dear Dr. Djuranovic,

Thank you for submitting your manuscript entitled "Role of the U11/U12 spliceosome gene ZCRB1 in Ciliogenesis and WNT Signaling" to Life Science Alliance. The manuscript was assessed by three expert reviewers, whose comments are appended to this letter.

The reviewers have all commented that the work is of interest and valuable to the community but have raised significant concerns that preclude publication in its current form. Based on their overall evaluation, we invite you to submit a revised manuscript.

As suggested by Reviewers 2 and 3 you must either provide evidence to support your claim of a direct relationship between Zcrb1, cilia, and Wnt or we suggest that you moderate your claims. Further, we bring your attention to concerns raised by all the reviewers on the need for a more accurate description of all phenotypes (cells or zebrafish) and need to better correlate effects of specific gene targeting with the observed phenotypes. For better correlation of phenotypes to genetic perturbation, we leave the method of validation to your choice. In the absence of this additional evidence you must temper your claims on the phenotypes.

Your revised manuscript must be improved for clarity, accuracy, completeness, and scientific rigour as primarily indicated by Reviewer 1 and as in minor comments by the other reviewers. Related to this please provide qPCR data with additional housekeeping genes (Reviewer 3) and quantification of Western blots in Figures 3C/E, S1B (Reviewers 1 and 2).

We agree that you must address the reviewers' concerns regarding more information on the RNA-seq dataset, in-depth analyses, and clarification of provided validation (Reviewer 1 (major point page7), Reviewer 2 (major point 2 on RNASeq) and Reviewer 3 (major point 1)).

I would be happy to discuss the revision in more detail via email or phone/videoconferencing. Please let me know which option you prefer, if any.

While you are revising your manuscript, please also attend to the below editorial points to help expedite the publication of your manuscript. Please direct any editorial questions to the journal office. When submitting the revision, please include a letter addressing the reviewers' comments point by point. While a rebuttal must respond to all points in some form, additional experiments to resolve these points, other than indicated above, will not be required.

Thank you for this interesting contribution to Life Science Alliance. We hope that the comments below will prove constructive as your work progresses, and we are looking forward to receiving your revised manuscript.

Sincerely,

Sarita Hebbar, PhD
Scientific Editor
Life Science Alliance
<http://www.lsajournal.org>

- A letter addressing the reviewers' comments point by point.
- An editable version of the final text (.DOC or .DOCX) is needed for copyediting (no PDFs).
- High-resolution figure, supplementary figure and video files uploaded as individual files: See our detailed guidelines for preparing your production-ready images, <https://www.life-science-alliance.org/authors>
- Summary blurb (enter in submission system): A short text summarizing in a single sentence the study (max. 200 characters including spaces). This text is used in conjunction with the titles of papers, hence should be informative and complementary to the title and running title. It should describe the context and significance of the findings for a general readership; it should be written in the present tense and refer to the work in the third person. Author names should not be mentioned.
- By submitting a revision, you attest that you are aware of our payment policies found here: <https://www.life-science-alliance.org/copyright-license-fee>

B. MANUSCRIPT ORGANIZATION AND FORMATTING:

Reviewer #1 (Comments to the Authors (Required)):

Pirzada and colleagues' work focused on the characterization of the function of ZCRB1, a protein that has been associated to the U11/U12 di-snRNP of the minor spliceosome, a machinery that is specifically involved in the excision of < 1% of introns in the human genome. To do so, they generated both cellular (HEK293) and zebrafish models, in which they performed RNA sequencing to uncover the mis-regulated genes. From these analyses, they pointed out that ciliary genes are mis-regulated, as well as genes involved in the Wnt signalling. To ascertain their findings, they analyse cilium formation in RPE1 cells, and explored the phenotype associated to *zcrb1* loss of function in zebrafish, which was reminiscent of an upregulation of Wnt pathway.

Even though these findings are of interest, they suffer from an overall lack of clarity and rigor of the manuscript (approximations, incorrect and/or misplaced references, typos, grammar...).

Other major points:

- Results, page 6. Authors confirmed the function of ZCRB1 as a component of the U12 snRNP, and showed that a significant number of genes associated to both spliceosomes are down-regulated, including those encoding proteins of U11 or U12 snRNP. In order to better describe the mechanisms downstream ZCRB1 loss of function, authors should investigate how ZCRB1 deficiency impacts the stability of U12 snRNP or U11/U12 di-snRNP.

Also, what is the rationale for verifying by northern blot the level of expression of all spliceosomal snRNAs?

- Results, page 7. Beyond gene expression, authors studied splicing events using rMATS bioinformatics tool. To my point of view, this dataset is not fully explored and could give further hints on the correlation/direct link between mis-splicing of U12 (or U2) introns and decrease of gene expression (via NMD for example).

They identified 3501 differential splicing events, of which 163 affect MIG transcripts. Of these 163 DS events, how many span the U12 intron? Of the remaining 3338 DS events, how many alter a U2-type intron splicing of a MIG (it has been shown for example that minor spliceosome deficiency can alter contiguous U2 introns of mis-spliced U12 introns)? or is it only the U2-type introns of U2 genes? This information will help appreciating whether loss of ZCRB1 function alters directly or indirectly major intron splicing.

How many DS events affect cilium/centrosome-related transcripts (consider all types of events rather than focusing on intron retention events only)? Is there a correlation between DS events and DE genes?

- Results, page 9. Gold standard for characterization of zebrafish model includes: sequence of gRNA target site to show indels,

RT-qPCR analysis to show the loss of expression of *zcrb1*, and if antibody is available, western blot analysis in rescued animals to validate the expression of human ZCRB1 protein.

- Results, page 10 & Figures 4E & 6D. Authors intend to demonstrate alteration of cilia in zebrafish. The experiments shown here are not convincing to me. First, the phenotype of defective gastrulation is not suggestive of ciliary defects in zebrafish. Second, the imaging of cilia raises questions. Which organ was analysed? What type of cilia is studied in Figures 4A & 6D? Also, imaging dying crispant embryos at 32 hpf (as seen in Figures 4C-D and 6A) is likely to result in absence of cilia... Authors should consider analysing cilia in embryos injected with lower doses of CRISPR/Cas RNP. Indeed, one could expect to observe ciliopathy-related phenotypes (body curvature, pronephric cysts, laterality defects,...) in *zcrb1* crispant.

- Statistics : Review the use of statistical tests. For instance, the low number of replicates in Fig. 1B (5 figures) rather justify the use of non parametric than parametric test.

Minor points :

- Introduction, page 3. Authors state that "loss of function mutations in minor spliceosome-associated proteins have implications for human disease", followed by "there are several congenital human diseases that have been shown to occur explicitly as a result of the dysregulation of one or more of the integral U12-dependent snRNP components". To illustrate these statements, authors mention mutations in U4atac snRNA, RNPC3 and CENATAC. These examples, except RNPC3, are just wrong : u4atac snRNA is a RNA and not a protein, and U4atac and CENATAC belong to the U4atac.U6atac.U5 tri snRNP (authors should read the review they mentioned, Norpaa et al., 2025, who is co-author of the manuscript...).

If authors want to give examples of proteins specific of U12 snRNP and associated to human diseases, they can cite ZRSR2 (PMID: 38158857) and SCNM1 (PMID: 36084634), in addition to RNPC3.

Of note, Li et al., 2023 is an inappropriate reference in this paragraph.

- Results, part 1. The title is incorrect : the protein ZCRB1 interacts with RNA - not ZCRB1 gene. Moreover, can authors be more specific : "interacts in an RNA dependent manner" with what?

- Figure S1A : Can authors provide the chromatograms for each site targeted by gRNA?

- Figure S1B : Given that protein loading in fluctuating (IB HSP70), a quantification of the WB would be expected to better appreciate the loss of ZCRB1 protein expression.

- Figure 1, legend and associated text in Results section : Authors should precise what "WT" means. Is it the parental (polyclonal) HEK293 cell line? Or one of WT clones issued from the CRISPR/Cas9 procedure? Also, it is unclear that graphs Fig. 1A and 1B show normalised/relative expression of ZCRB1 to GOLGA5 protein/transcript.

- Results, pages 5 and 10. Precise in the text that a bulk short-reads RNAseq analysis was done; precise the number of technical replicates; show the heteroscedasticity in supplementary figure; indicate in the text the fold change that was chosen. In M&M, authors indicate that total RNA was depleted from ribosomal RNA, and then mentioned mRNA for reverse transcription: does it mean that a purification of polyA+ mRNA has been done? This information needs to be clearly stated, in the Results section too, as it may explain why RNAseq poorly captured snRNA genes (transcripts not polyadenylated).

- Results, page 6 : The following sentence is wrong : "including snRNA genes belonging to both major and minor spliceosomes, such as U1 and U12 snRNP". Authors seem to confuse snRNA and snRNP. Among the detected snRNA genes are RNU4.2, RNU4.1, RNU5F.1, RNU2.63P and few RNU1 variants; there are no RNU genes belonging specifically to minor spliceosome, especially not RNU12 snRNA. Please, clarify the sentence here.

- Tables S1 and S3. Add a legend to the tables. What is "Cl.L", Cl.R", "t", "B" ? In S1, add a tab detailing GO term analysis of 164 misexpressed MIGs.

- In the text, Figure S4 is mentioned before Figure S3; please, reorder supplementary figures.

- Figure 2C. Specify in the legend that the schema summarizes the function of the cilium/centrosome-related (U2 and U12) genes that are DE in ZCRB1-deficient cells. Why only 76 of the 116 DE genes are present in the figure? A color code for up-/down-regulated, and MIG genes (bold) would be appreciated in this figure.

In the text, page 7, specify which database of cilium/centrosome-related genes was used to screen DE genes.

- Figures 2D-2E: Precise whether the depicted introns are U12 or not. Add a schema of the splicing event that correspond to each amplicon seen by RT-PCR. On the example of RABL2B, it is rather an alternative 5' splicing site than an intron retention (see Table S2).

- Figures 3C & 3E: Western blots should be quantified. Of note, increase of pLRP6 may just be a result of increase of total LRP6 protein level (analyse ratio pLRP6/total LRP6 to conclude on phosphorylation level).

- Figure 4A. Please draw the RRM and ZF domains on the protein sequence to better appreciate the high degree of identity between human and zebrafish proteins.

- Results, page 10. Authors should precise at what stage the RNAseq analysis in zebrafish was performed. Also, the sentence "Many of these DE genes fell into the same categories as those identified in the heterozygous human ZCRB1 mutant dataset, belonging to BBSome, IFT,..." is superfluous here, since ciliary/centrosomal gene analysis is described few lines below.

- Discussion, page 12. "we did not observe significant downregulation of U12 and U1 snRNA seen in RNAseq data". U12 snRNA does not appear in the RNAseq dataset. Please correct the sentence here.

- M&M : RPE1 cell culture is missing.

Reviewer #2 (Comments to the Authors (Required)):

1. Summary of the findings of the manuscript and their importance:

The manuscript by Rehman Pirzada, Sergej Djuranovic, and colleagues examines the role of the spliceosome gene ZCRB1. Using gene editing, siRNA, transcriptomic, and cell biology approaches in vertebrate cell lines and zebrafish embryos, the authors report prominent deregulation of cilia- and Wnt pathway-related genes in models with disrupted ZCRB1 function. The key findings, which link ZCRB1 to the regulation of proper cilia biogenesis and the tuning of Wnt/ β -catenin signaling, are novel and interesting, yet somewhat descriptive. Still, the manuscript could serve as a useful resource for readers seeking an initial insight into how spliceosome defects may translate into deregulation of developmental pathways. The final part of the manuscript on the causal relationship between the observed defects in ciliogenesis and Wnt signaling feels very premature, though.

2. Key findings of the manuscript:

ZCRB1 mutant generation in HEK293 cells and analysis of differentially expressed genes by RNA Seq.

The relevant results (Figs. 1-2 and related Supplementary figures and tables) are well documented and mostly convincing; the conclusions are sound and consistent with the presented data. The enrichment of cilia-related genes among the small intron-containing deregulated genes is intriguing. I appreciate the effort to validate the intron retention suggested by the RNA-seq dataset using RT-PCR (Fig. 2D-E and Supplementary Fig. 6B). While the gain of extra sequence is clearly visible for the RABL2B and IFT88 transcripts, it is not clear to me how the presence of multiple bands in the WT sample versus one prominent band in the "54" sample serves as evidence of intron retention in the CCDC28B transcript (Fig. 2E). Please clarify.

Depletion of ZCRB1 in RPE1 cells impairs ciliogenesis and elevates Wnt/ β -catenin signaling.

In Fig. 3, the authors examine the effect of ZCRB1 depletion (siRNA) on cilia formation in RPE1 cells. They report an effect on primary cilia formation in ZCRB1 siRNA-transfected cells, as well as changes in the protein levels of IFT88 and other cilia/centrosome components, in line with their RNA-seq data. Importantly, the included Western blot data show that ZCRB1 protein levels are efficiently depleted (Fig. 3C). This part is quite clear, provided that the authors add information on the number of cells examined for each condition/experiment when quantifying the incidence of ciliogenesis (to MM or figure legends). The data in Fig. 3D-E suggest upregulation of the Wnt/ β -catenin signaling pathway (higher signal for LRP6/pLRP6 and AXIN2). However, it is not clear from the text (Results, figure legends, or Methods) whether the Western blot analysis represents a single experiment (the presented blot) or whether multiple independent experiments were performed with similar outcomes to demonstrate reproducibility.

Zcrb1 is required for zebrafish embryo development.

The authors used F0 crispants to examine the impact of Zcrb1 ablation on zebrafish embryo development. As the use of crispants comes with several tradeoffs (e.g., chimerism and off-target activity), I appreciate that the authors included a rescue experiment with human ZCRB1 mRNA, which indeed ameliorates the gastrulation and body axis defects (Fig. 4D) as well as the reduced number of cilia (Fig. 4E-F) observed in the crispants. To better correlate the performed gene editing with the penetrance of the observed phenotypes, could the authors provide sequencing-based analyses (e.g., Sanger sequencing with ICE analysis or NGS) to illustrate the efficiency of the gRNAs used in these experiments?

Impaired zcrb1 function leads to increased Wnt signaling downstream of cilia loss in the zebrafish.

The RNA-seq analyses of zebrafish crispants suggest an upregulation of the Wnt signaling pathway, in line with transcriptomic data from the mutant HEK cell lines. However, the proposed causality—namely, that the observed defects in Wnt signaling in zcrb1 crispants are caused by defective ciliogenesis—is clearly the weakest part of the manuscript. The experiment using a tankyrase inhibitor reports mitigation of gastrulation and body axis defects, yet no rescue of the defective ciliogenesis. This suggests that elevated Wnt/ β -catenin signaling is likely the major pathway responsible for the observed body axis defects, while it is not involved in the ciliogenesis phenotype. Importantly, however, these data provide no evidence that Wnt pathway deregulation occurs downstream of defective cilia formation.

The effects of zcrb1 on the Wnt pathway could be entirely independent of its effects on cilia, as loss of zcrb1 appears to deregulate numerous Wnt pathway components, including pathway ligands. Indeed, inspection of the list of minor intron-

deregulated genes (Table S1) reveals RNF220, which is upregulated in *zcrb1* mutants and encodes an E3 ligase implicated in the stabilization of β -catenin and facilitation of Wnt/ β -catenin signaling (PMID: 25266658). This could, in principle, also explain the observed Wnt pathway upregulation. Alternatively, several IFT components have been proposed to regulate Wnt/ β -catenin signaling independently of their role in cilia (PMID: 30546012; PMID: 38870008). Moreover, the relationship between Wnt signaling and cilia is considerably more controversial than the relatively straightforward cilia-Hedgehog pathway link (e.g., PMID: 28741966; PMID: 38043953), with published evidence showing that ablation of key regulators of ciliogenesis does not affect Wnt/ β -catenin pathway activation in zebrafish (PMID: 19700616).

It is therefore vital that the authors revise the manuscript accordingly to reflect these points. If they wish to retain their conclusion regarding a causal *zcrb1*-cilia-Wnt relationship, they will need to provide substantial additional evidence—for example, by ablating cilia formation through means other than *zcrb1* disruption and subsequently assessing Wnt pathway activation in such models, while excluding alternative mechanisms (such as cilia-independent roles of IFT proteins or direct regulation of Wnt components expression by *zcrb1*). In addition, the authors should carefully discuss any discrepancies between their findings and the existing literature.

3. Additional recommendations:

It is not clear why Wnt5a is referred to in the text (Results, section related to Fig. 5) as a Wnt pathway inhibitor, as it is well established to be a ligand capable of activating multiple signaling pathways depending on the available receptors (PMID: 22771246).

In addition, the statement in the Introduction that "WNT signaling in particular can both transmit..." does not take into account the literature arguing against a direct role of cilia in Wnt pathway signal transduction. Please revise this statement accordingly.

Of note, the lack of any page/line numbering in the manuscript made its evaluation a bit more tedious than it needed to be.

Reviewer #3 (Comments to the Authors (Required)):

Pirzada et al demonstrates reduction in ZCRB1, a component of the minor spliceosome, impacts splicing and expression of minor intron containing genes (MIGs) as would be expected. The novel aspect of this study identifies an enrichment of MIGs essential for ciliogenesis that are disrupted leading to impaired cilia formation, increased WNT signaling and developmental patterning defects. These defects can be rescued by Wnt inhibition but not cilia restoration. The use of both human cell line studies and in vivo zebrafish models support the conservation of these observations in animal development.

However I believe there are two major points that need to be addressed to strengthen the paper and support the conclusions drawn.

Major points:

1. Further analysis of RNAseq to support the statement that ciliogenesis MIGs are preferentially impacted by ZCRB1 depletion.

The HEK293 RNAseq analysis which forms the basis for investigating ciliogenesis is confounded by a global effect on dysregulation of RNA metabolism and splicing components (Fig 1C & 1D). The phenotypes observed may be a consequence of the cellular stress from the reduction in RNP complexes. This limitation should be acknowledged.

The RNAseq dataset could be further analysed to interegate all the MIGs and control for expression, intron length, splice site sequence etc. Are cilia MIGs specifically impacted relative to other MIGs? Are cilia MIGs specifically impacted relative to other cilia (NON-MIG) genes? In addition the RT-PCR confirmation of MIG splicing changes should be strengthened/expanded (currently figure 2E is not convincing - see point 4 below) to a larger panel of cilia MIG genes.

The RNAseq should be compared to other published datasets for minor splicing depleted cell lines? Is the enrichment in impaired spicing of ciliogenesis MIGs specific to ZCRB1 depletion? Is this effect specific to HEK293 cells? Are there datasets of cells more appropriate to studied ciliogenesis eg. RPE-1 cells?

2. Confirmation of zebrafish development and embryonic patterning defects by ZCRB1 depletion should be extended

Currently figure 4C-D provides evidence for essentiality of ZCRB1 for zebrafish development but this is all based upon F0 CRISPR mosaic embryos generated from a single gRNA.

While the mRNA rescue experiments support the claim ZCRB1 is responsible for the developmental defects the potential for off target effects remain and independent guides or stable mutant lines are required to confirm the observation.

Minor points:

1. Should be consistant in terminology and use minor introns or U12-type introns throughout and not switch between both.
2. Large spread between biological replicates in RT-qPCR data presented in Figure 1B. As these are clonal cell lines with

heterozygous alleles I would not expect the variation to be so big. This may be confounded by the global dysregulation of RNA metabolism and should be commented on/addressed. Data could be strengthened by using more than one housekeeper gene to normalise cDNA levels. Ideal housekeeper gene candidates could be identified from RNAseq dataset.

3. Label x-axis of figure 1D.

4. Figure 2E (&2D) - Unclear if there is a specific amplicon in WT control sample for CCDC28B. Not a strong example to support data. More detail in legend or on figure would help - Label the minor intron and position of primers for the RT-PCR on the gene schematic?

5. Care should be taken to not over claim that ZCRB1 is an essential gene for cellular fitness and survival in human cells. Data here suggests this for HEK293 cells but complete loss of ZCRB1 was not tested in any other human cell lines. Also absence of recovering null alleles is not conclusive evidence of essentiality. Consider altering the text to reflect this.

6. The cilia reduction in human RPE-1 cells by ZCRB1 siRNA is compelling (figure 3A-C) but no genetic rescue was performed (siRNA resistant ZCRB1) to confirm phenotype was due to on target siRNA activity. This should be performed or an alternative ZCRB1 depletion technique such as CRISPR could be utilised as an orthogonal method to support ZCRB1 specificity.

7. The upregulation of WNT signaling components (figure 3E) is not quantitative, replicates and densitometry would strengthen data. Acknowledge the phenotype could be a secondary effect of the cell stress caused by the global dysregulation of RNA metabolism seen in ZCRB1 depleted cells (see major point 1). The most direct functional read out for WNT pathway activation would be beta catenin staining/nuclear translocation or TCF/LEF transcriptional activity. This data from human cells would complement the zebrafish studies.

8. Rescue of the upregulated WNT signaling components in the ZCRB1 HEK293 clones (figure 3E) or reproducing the observation in a second human cell line is required to strengthen the claim that loss of ZCRB1 leads to upregulation of WNT signaling in human cells.

9. Should reference new paper PMID: 41258964 (Nov. 2025) which looks at ZCRB1 disruption in liver cancer.

We would like to thank the reviewers for the thorough and positive review of our manuscript. Below, we include our point-by-point responses to the concerns raised and highlight the changes we have made to our manuscript to meet the expectations for publication.

Reviewer #1:

Pirzada and colleagues' work focused on the characterization of the function of ZCRB1, a protein that has been associated to the U11/U12 di-snRNP of the minor spliceosome, a machinery that is specifically involved in the excision of < 1% of introns in the human genome. To do so, they generated both cellular (HEK293) and zebrafish models, in which they performed RNA sequencing to uncover the mis-regulated genes. From these analyses, they pointed out that ciliary genes are mis-regulated, as well as genes involved in the Wnt signalling. To ascertain their findings, they analyse cilium formation in RPE1 cells, and explored the phenotype associated to zcrb1 loss of function in zebrafish, which was reminiscent of an upregulation of Wnt pathway.

Even though these findings are of interest, they suffer from an overall lack of clarity and rigor of the manuscript (approximations, incorrect and/or misplaced references, typos, grammar...).

Response: We thank the reviewer for the deep read of our work and for the valuable comments. We have now changed multiple figures and references, increased clarity in the text, and added additional data to support role of ZCRB1 in the minor spliceosome, splicing of cilia regulated genes, and impact on WNT signaling. We will detail these additions below.

Results, page 6. Authors confirmed the function of ZCRB1 as a component of the U12 snRNP, and showed that a significant number of genes associated to both spliceosomes are down-regulated, including those encoding proteins of U11 or U12 snRNP. ***In order to better describe the mechanisms downstream ZCRB1 loss of function, authors should investigate how ZCRB1 deficiency impacts the stability of U12 snRNP or U11/U12 di-snRNP. Also, what is the rationale for verifying by northern blot the level of expression of all spliceosomal snRNAs?***

Response: RNA-Seq analyses across ZCRB1 cell lines (with Ribo Zero applied and without polyA+ selection) indicated changes in major and minor spliceosome snRNA levels (Table 1). We wanted to validate these results by northern blot, as structure and sequence modifications of snRNAs are reported to affect sequencing results and lead to differences by RNA-Seq that might not exist. This previously reported artifact is what prompted us to investigate the expression of all spliceosome-associated snRNA levels by northern blot. This analysis showed that expression of the snRNAs tested was largely unaffected, a point we felt warranted mentioning in the text, as it provides additional context for the interpretation of our data.

Additionally, as per reviewer's suggestion, we investigated the stability of U11/U12 di-snRNP specific proteins, i.e. RNPC3 and PDCD7, in ZCRB1-heterozygous clone 54 cells. Both proteins were unaffected compared to parental HEK293-WT control cells, similar to

snRNA levels shown in the manuscript (the RNPC3 and PDCD7 data is not mentioned in our manuscript but shown below for the reviewers; panel **A**). We also performed ZCRB1-IP experiments in *ZCRB1*-heterozygous clone 54 to investigate effects on levels of U11/U12 di-snRNP RNPC3. The results again revealed no significant changes to the levels of RNPC3 being pulled down in the *ZCRB1*-heterozygous condition (results attached below but not shown in the manuscript; panel **B**) Taken together, these results suggest decreased levels of *ZCRB1* induce the splicing defects without affecting the stability of other minor snRNPs or snRNAs. Future work will employ the gold standard methods of native complex analysis, assessing the respective snRNA half lives in an inducible sh-*ZCRB1* to yield more information regarding effects on stability.

Results, page 7. Beyond gene expression, authors studied splicing events using rMATS bioinformatics tool. To my point of view, **this dataset is not fully explored and could give further hints on the correlation/direct link between mis-splicing of U12 (or U2) introns and decrease of gene expression (via NMD for example).**

Response: We fully agree with the reviewer; however, full dissection of this dataset is complex, and parsing the role of NMD versus other mechanistic regulators of gene expression became too large for the scope of this manuscript. To address this point, we have included additional examples and validation of mis-splicing events following loss of *ZCRB1* and have mentioned these future considerations in our discussion (Fig. 2D & line 385-389).

They identified 3501 differential splicing events, of which 163 affect *MIG* transcripts. Of these 163 DS events, **how many span the U12 intron?** Of the remaining 3338 DS events, **how many alter a U2-type intron splicing of a MIG** (it has been shown for example that minor spliceosome deficiency can alter contiguous U2 introns of mis-spliced U12 introns)? **or is it only the U2-type introns of U2 genes?** This information will help appreciating whether loss of *ZCRB1* function alters directly or indirectly major intron splicing.

Response: In our dataset, we identified 3,502 differential splicing events, of which 165

affect minor intron-containing genes (MIGs) with U12-type introns. Most of these events occur in U2-type introns of MIGs, with the exception of *MAPK12* and *CCDC28B*, where we detect inclusion of the U12-type intron itself; *CCDC28B* is a well-established regulator of ciliogenesis. Retained intron (RI) events are significantly enriched for MIGs, as determined by Fisher's exact test, with an odds ratio of 1.82 and a false discovery rate (FDR)-adjusted p-value of 0.034, indicating that MIGs are preferentially impacted in these events. This enrichment likely reflects the interdependency of minor and major introns within the same transcript: because U12-type introns are typically rate-limiting for splicing, loss of ZCRB1 may perturb splicing of adjacent U2-type introns, resulting in broader co-transcriptional splicing defects. Overall, these results suggest that ZCRB1 deficiency has a global impact on splicing in minor intron-containing genes, beyond direct effects on U12-type introns (Table S2).

How many DS events affect cilium/centrosome-related transcripts (consider all types of events rather than focusing on intron retention events only)? Is there a correlation between DS events and DE genes?

Response: Across all differential splicing (DS) event types, we identified 128 events impacting transcripts associated with primary cilia and the centrosome. Of the 116 cilia-related genes showing differential expression (DE), 29 unique genes were both alternatively spliced and expressed. Notably, more than half of these (18/29) exhibited decreased expression. These data suggest that a substantial subset of cilia/centrosome genes is subject to both transcriptional and post-transcriptional regulation, as a result of loss of ZCRB1 function (Table S2).

Results, page 9. Gold standard for characterization of zebrafish model includes: **sequence of gRNA target site to show indels, RT-qPCR analysis to show the loss of expression of *zcrb1*, and if antibody is available, western blot analysis in rescued animals to validate the expression of human ZCRB1 protein.**

Response: We appreciate the goal to quantify the level of *zcrb1* disruption in the zebrafish model; however, these experiments are complicated by several factors. First, we lack an antibody that we currently feel confident is recognizing Zcrb1 protein in the zebrafish. Second, all the injection studies presented in the manuscript are mosaic, with the embryos all harboring some edited and some WT *zcrb1* transcript. This causes the RT-qPCR analysis to be extremely variable between single embryos and impossible to carry out on pooled samples of embryos.

To address these points and validate the phenotypic specificity of our guide RNA targeting, we followed the community standards laid out in Stainier, et al. 'Guidelines for morpholino use in zebrafish' and applied them to our crisprant guide RNA injection models. First, we carried out phenotyping studies with multiple guide RNAs to different regions of the *zcrb1* transcript. 3/4 of the guide RNAs showed consistent phenotypes, which we now show in Supplemental Figure 9. Second, as shown in the original submission, we carried out injection of human WT ZCRB1 mRNA to confirm phenotypic rescue and specificity of these phenotypes to the injected *zcrb1* guide RNA. This data remains presented in Figure 4. Finally, as recommended and requested, we show targeted DNA sequencing analysis and gDNA fragment analysis of the genomic region targeted by the guide RNA. These

analyses show clear generation of indels at the predicted guide RNA cut site, and this data is now included as Supplementary Figure 10. We hope this raises the reviewer's confidence in our approach.

Results, page 10 & Figures 4E & 6D. Authors intend to demonstrate alteration of cilia in zebrafish. The experiments shown here are not convincing to me. **First, the phenotype of defective gastrulation is not suggestive of ciliary defects in zebrafish.**

Response: We agree with the reviewer that gastrulation is not necessarily linked to a defective cilia phenotype (laterality defects, cysts, etc.). In this case, the gastrulation defects—which we show are caused by dysregulated Wnt signaling—likely predate potential cilia-mediated defects. In order to show disrupted cilia in this model, we imaged the Arl13b-GFP transgenic line in living *zcrb1* crispant embryos (i.e. embryo's retaining a heartbeat) showing the presence of significantly fewer cilia.

Second, the imaging of cilia raises questions. **Which organ was analysed? What type of cilia is studied in Figures 4A & 6D? Also, imaging dying crispant embryos at 32 hpf (as seen in Figures 4C-D and 6A) is likely to result in absence of cilia.**

Authors should consider analysing cilia in embryos injected with lower doses of CRISPR/Cas RNP. Indeed, one could expect to observe ciliopathy-related phenotypes (body curvature, pronephric cysts, laterality defects,...) in *zcrb1* crispant.

Response: Our analysis is focused on primary cilia. Z-stack images were acquired capturing the medial through lateral planes of the embryo (using the dorsal aorta as a marker, magenta in the images in Fig. 4D), which encompassed the skin and skeletal muscle-associated cilia present on the embryo's right side. Across these studies, only embryos that retained a heartbeat were selected for imaging (additional examples are now shown in Fig. S11 for the WNT reporter, but the same logic was applied in the cilia studies). We have updated the text with this information for clarity.

Statistics: Review the use of statistical tests. For instance, the low number of replicates in Fig. 1B (5 figures) rather justify the use of non parametric than parametric test.

Response: Although the sample sizes in the indicated panels are small, the data showed no evidence of strong deviations from normality, unequal variance, or outliers. Normality was assessed using the Shapiro-Wilk test, which is commonly used for small sample sizes, and no significant deviations from normality were detected. Because the groups have equal sample sizes and similar variances, one-way ANOVA is considered robust to moderate deviations from normality. Therefore, we used parametric tests for statistical analysis in these panels.

Minor-points:

Introduction, page 3. Authors state that "loss of function mutations in minor spliceosome-associated proteins have implications for human disease", followed by "there are several congenital human diseases that have been shown to occur explicitly as a result of the

dysregulation of one or more of the integral U12-dependent snRNP components". To illustrate these statements, authors mention mutations in U4atac snRNA, RNPC3 and CENATAC. These examples, except RNPC3, are just wrong : u4atac snRNA is a RNA and not a protein, and U4atac and CENATAC belong to the U4atac.U6atac.U5 tri snRNP (authors should read the review they mentioned, Norpaa et al., 2025, who is co-author of the manuscript...).

If authors want to give examples of proteins specific of U12 snRNP and associated to human diseases, they can cite ZRSR2 (PMID: 38158857) and SCNM1 (PMID: 36084634), in addition to RNPC3. Of note, Li et al., 2023 is an inappropriate reference in this paragraph.

Response: Universally, snRNP components includes both snRNAs and associated proteins. We have modified the text to improve our clarity on this point. We have also updated the reference as requested (line 103).

Results, part 1. **The title is incorrect:** the protein ZCRB1 interacts with RNA - not ZCRB1 gene. Moreover, can authors be more specific: "interacts in an RNA dependent manner" with what?

Response: We have updated the title to "ZCRB1 is required for cellular homeostasis and interacts with minor spliceosome proteins in an RNA-dependent manner" line 140-141.

Figure S1A: Can authors provide the chromatograms for each site targeted by gRNA?

Response: As an alternative to Sanger sequencing chromatograms, CRISPR target site validation was performed using whole-genome sequencing (WGS). Visualization of CRAM file alignments in Integrative Genomics Viewer (IGV) confirmed the expected site of integration and revealed indels at the target locus, which were absent in control samples and the reference genome, supporting successful genome editing. This data has been included in our revised manuscript Fig S1 B.

Figure S1B: Given that protein loading in fluctuating (IB HSP70), a quantification of the WB would be expected to better appreciate the loss of ZCRB1 protein expression.

Response: We have quantified the blot, and the data has been included in Figure S2B.

Figure 1, legend and associated text in Results section: **Authors should precise what "WT" means. Is it the parental (polyclonal) HEK293 cell line? Or one of WT clones issued from the CRISPR/Cas9 procedure? Also, it is unclear that graphs Fig. 1A and 1B show normalised/relative expression of ZCRB1 to GOLGA5**

protein/transcript.

Response: WT meant the parental HEK293 cells and this information has been updated (line 149-150). Fig 1A shows the ZCRB1 protein normalization relative to GOLGA5 (line 151) and Fig 1B shows ZCRB1 transcript normalization to GOLGA5 (line 152). This information has been updated in the respective figure legends as well.

Results, pages 5 and 10. Precise in the text that a bulk short-reads RNAseq analysis was done; precise the number of technical replicates; show the heteroscedasticity in supplementary figure; indicate in the text the fold change that was chosen.

Response: In order to show that lack of ZCRB1 induces similar genomic effects in different CRISPR-Cas9 clones, we did RNA-seq analyses on three independent HEK293 clones that underwent CRISPR-Cas9 editing but were confirmed by whole-genome sequencing to remain wild-type at the ZCRB1 locus, and four different ZCRB1 heterozygote mutant cell lines. The overall differences as well as P-values were calculated for each individual analysis, and we cross referenced the lists to focus only on genes that were affected across all ZCRB1 heterozygous clones. Log2FC <-0.58 or > 0.58 was chosen; this information is updated in line 703. P-values were adjusted for multiple testing using the Benjamini-Hochberg false discovery rate (FDR). While we do not explicitly show a figure for the absence of heteroscedasticity, the differential expression pipeline we used, which included TMM normalization in EdgeR, Limma voom With Quality Weights for mean-variance modeling, and application of sample-specific quality weights, explicitly accounts for heteroscedasticity by stabilizing variance across expression levels.

In M&M, authors indicate that total RNA was depleted from ribosomal RNA, and then mentioned mRNA for reverse transcription: **does it mean that a purification of polyA+ mRNA has been done? This information needs to be clearly stated, in the Results section too, as it may explain why RNAseq poorly captured snRNA genes (transcripts not polyadenylated).**

Response: Because the Ribosomal RNA (rRNA) constitute nearly 80-90% of total cellular RNA we used RiboErase to deplete rRNAs, however no enrichment was done towards polyA containing mRNAs as we wanted to see effects on other RNA species (please see answer above for Northern blots). In that pool the capturing of RNA depends on their abundance. This has been updated in M&M line 681-682.

Results, page 6: The following sentence is wrong: "including snRNA genes belonging to both major and minor spliceosomes, such as U1 and U12 snRNP". Authors seem to confuse snRNA and snRNP. Among the detected snRNA genes are RNU4.2, RNU4.1, RNU5F.1, RNU2.63P and few RNU1 variants; there are no RNU genes belonging specifically to minor spliceosome, especially not RNU12 snRNA. **Please, clarify the sentence here.**

Response: The discrepancy between snRNP and snRNA (line 168-183) has been updated to improve clarity. We however would like to point out that our RNA sequencing including the major spliceosome RNUs also shows a -3.51 fold decrease in U12 snRNA

(Table 1S, sheet 4 row 378, and sheet 6 row 56). The original analysis pipeline has assigned it being lncRNA instead of snRNA.

Tables S1 and S3. Add a legend to the tables. What is "Cl.L", Cl.R", "t", "B" ? In S1, add a tab detailing GO term analysis of 164 misexpressed MIGs.

Response: An abbreviations table has been incorporated in content sheet (sheet 1) of table S1. GO information of 165 differentially expressed MIGs has been added as sheet 8 in table S1.

In the text, Figure S4 is mentioned before Figure S3; please, reorder supplementary figures.

Response: This has been rectified.

Figure 2C. Specify in the legend that the schema summarizes the function of the cilium/centrosome-related (U2 and U12) genes that are DE in ZCRB1-deficient cells. Why only 76 of the 116 DE genes are present in the figure? A color code for up/down-regulated, and MIG genes (bold) would be appreciated in this figure.

Response: Thank you for this suggestion, the panel has been updated.

In the text, page 7, specify which database of cilium/centrosome-related genes was used to screen DE genes.

Response: Cilia-associated genes were annotated using KEGG BRITe functional classifications, specifically genes assigned to the "cilium and associated proteins" category.

Figures 2D-2E: Precise whether the depicted introns are U12 or not. Add a schema of the splicing event that correspond to each amplicon seen by RT-PCR. On the example of RABL2B, it is rather an alternative 5' splicing site than an intron retention (see Table S2).

Response: These points have been addressed in respective figures and legends. Alternate 5' start site events were enriched among cilia-associated genes (odds ratio = 1.7; one-sided Fisher's exact test, $p = 0.0249$). For this reason, we also depict the alternate 5' splice usage in RABL2B, highlighting a representative cilia gene affected by this splicing category.

Figures 3C & 3E: Western blots should be quantified. Of note, increase of pLRP6 may just be a result of increase of total LRP6 protein level (analyse ratio pLRP6/total LRP6 to conclude on phosphorylation level).

Response: The blots have been quantified and the data included in Figure S8 A,B.

Figure 4A. Please draw the RRM and ZF domains on the protein sequence to better appreciate the high degree of identity between human and zebrafish proteins.

Response: This has been updated Fig. 4A.

Results, page 10. **Authors should precise at what stage the RNAseq analysis in zebrafish was performed. Also, the sentence "Many of these DE genes fell into the same categories as those identified in the heterozygous human ZCRB1 mutant dataset, belonging to BBSome, IFT,..." is superfluous here**, since ciliary/centrosomal gene analysis is described few lines below.

Response: We have added that the RNA-seq was done at 28 hpf and removed the suggested sentence.

Discussion, page 12. "we did not observe significant downregulation of U12 and U1 snRNA seen in RNAseq data". **U12 snRNA does not appear in the RNAseq dataset. Please correct the sentence here.**

Response: U12 is labeled in Table 1 (Sheet 4 row 378, and sheet 6 row 56) as lncRNA and not as snRNA and it shows differences between heterozygote and wild type cells. The same is the case for pseudogenes of U1 and U1 snRNA. Our statement in the discussion is meant to refer to the Northern blot analyses and has been clarified.

M&M: RPE1 cell culture is missing.

Response: This has been updated (line 483).

Reviewer #2 (Comments to the Authors (Required)):

Summary of the findings of the manuscript and their importance:

The manuscript by Rehman Pirzada, Sergej Djuranovic, and colleagues examines the role of the spliceosome gene ZCRB1. Using gene editing, siRNA, transcriptomic, and cell biology approaches in vertebrate cell lines and zebrafish embryos, the authors report prominent deregulation of cilia- and Wnt pathway-related genes in models with disrupted ZCRB1 function. The key findings, which link ZCRB1 to the regulation of proper cilia biogenesis and the tuning of Wnt/ β -catenin signaling, are novel and interesting, yet somewhat descriptive. Still, the manuscript could serve as a useful resource for readers seeking an initial insight into how spliceosome defects may translate into deregulation of developmental pathways. The final part of the manuscript on the causal relationship between the observed defects in ciliogenesis and Wnt signaling feels very premature, though.

Key findings of the manuscript:

ZCRB1 mutant generation in HEK293 cells and analysis of differentially expressed genes by RNA Seq. The relevant results (Figs. 1-2 and related Supplementary figures and tables) are well documented and mostly convincing; the conclusions are sound and consistent with the presented data. The enrichment of cilia-related genes among the small intron-containing deregulated genes is intriguing. I appreciate the effort to validate the intron retention suggested by the RNA-seq dataset using RT-PCR (Fig. 2D-E and Supplementary Fig. 6B).

While the gain of extra sequence is clearly visible for the RABL2B and IFT88 transcripts, it is not clear to me how the presence of multiple bands in the WT sample versus one prominent band in the "54" sample serves as evidence of intron retention in the CCDC28B transcript (Fig. 2E). Please clarify.

Response: This text has been updated to improve clarity. Since *CCDC28B* shows both retention of a minor intron and mutually exclusive exon alterations, we amplified its complete coding sequence to capture all the splicing alterations. Due to decreased levels of *CCDC28B* transcript compared to WT—perhaps happening due to increased turnover of its altered mRNA—we are unable to effectively capture the intron retention event. We also acknowledge that it doesn't show a prominent intron retention event so we have moved it to Supplementary Figure 6 B and the respective figure legend has been updated accordingly. We also acknowledge the limitation of sequencing reads that might affect the readout of our rMATs analysis for some of the minor intron-containing ciliary genes.

Depletion of ZCRB1 in RPE1 cells impairs ciliogenesis and elevates Wnt/beta-catenin signaling.

In Fig. 3, the authors examine the effect of ZCRB1 depletion (siRNA) on cilia formation in RPE1 cells. They report an effect on primary cilia formation in ZCRB1 siRNA-transfected cells, as well as changes in the protein levels of IFT88 and other cilia/centrosome components, in line with their RNA-seq data. Importantly, the included Western blot data show that ZCRB1 protein levels are efficiently depleted (Fig. 3C). ***This part is quite clear, provided that the authors add information on the number of cells examined for each condition/experiment when quantifying the incidence of ciliogenesis (to MM or figure legends).***

Response: We analyzed at least 50 cells per sample per replicate. We have updated the respective figure legend (Fig. 3B).

The data in Fig. 3D-E suggest upregulation of the Wnt/ β -catenin signaling pathway (higher signal for LRP6/pLRP6 and AXIN2). However, ***it is not clear from the text (Results, figure legends, or Methods) whether the Western blot analysis represents a single experiment (the presented blot) or whether multiple independent experiments were performed with similar outcomes to demonstrate reproducibility.***

Response: This point has been clarified in the text. We carried out multiple independent experiments for all of our included analysis. Please refer to Fig S8 A,B and its respective legend.

Zcrb1 is required for zebrafish embryo development.

The authors used F0 crispants to examine the impact of Zcrb1 ablation on zebrafish embryo development. As the use of crispants comes with several tradeoffs (e.g., chimerism and off-target activity), I appreciate that the authors included a rescue experiment with human ZCRB1 mRNA, which indeed ameliorates the gastrulation and body axis defects (Fig. 4D) as well as the reduced number of cilia (Fig. 4E-F) observed in the crispants. ***To better correlate the performed gene editing with the penetrance of the observed phenotypes, could the authors provide sequencing-based***

analyses (e.g., Sanger sequencing with ICE analysis or NGS) to illustrate the efficiency of the gRNAs used in these experiments?

Response: As requested, we now show targeted DNA sequencing analysis and gDNA fragment analysis of the genomic region targeted by our guide RNAs. These analyses confirm the generation of indels at the predicted guide RNA cut site and are now included as Supplementary Figure 10. Additionally, we carried out phenotyping studies with multiple guide RNAs targeting distinct regions of the *zcrb1* transcript. 3/4 of the guide RNAs showed consistent phenotypes, which we now show in Supplemental Figure 9.

Impaired *zcrb1* function leads to increased Wnt signaling downstream of cilia loss in the zebrafish.

The RNA-seq analyses of zebrafish crispants suggest an upregulation of the Wnt signaling pathway, in line with transcriptomic data from the mutant HEK cell lines. However, the proposed causality—namely, that the observed defects in Wnt signaling in *zcrb1* crispants are caused by defective ciliogenesis—is clearly the weakest part of the manuscript. The experiment using a tankyrase inhibitor reports mitigation of gastrulation and body axis defects, yet no rescue of the defective ciliogenesis. This suggests that elevated Wnt/ β -catenin signaling is likely the major pathway responsible for the observed body axis defects, while it is not involved in the ciliogenesis phenotype. Importantly, **however, these data provide no evidence that Wnt pathway deregulation occurs downstream of defective cilia formation.**

The effects of *zcrb1* on the Wnt pathway could be entirely independent of its effects on cilia, as loss of *zcrb1* appears to deregulate numerous Wnt pathway components, including pathway ligands. Indeed, inspection of the list of minor intron-deregulated genes (Table S1) reveals RNF220, which is upregulated in *zcrb1* mutants and encodes an E3 ligase implicated in the stabilization of β -catenin and facilitation of Wnt/ β -catenin signaling (PMID: 25266658). This could, in principle, also explain the observed Wnt pathway upregulation. Alternatively, several IFT components have been proposed to regulate Wnt/ β -catenin signaling independently of their role in cilia (PMID: 30546012; PMID: 38870008). Moreover, the relationship between Wnt signaling and cilia is considerably more controversial than the relatively straightforward cilia-Hedgehog pathway link (e.g., PMID: 28741966; PMID: 38043953), with published evidence showing that ablation of key regulators of ciliogenesis does not affect Wnt/ β -catenin pathway activation in zebrafish (PMID: 19700616).

It is therefore vital that the authors revise the manuscript accordingly to reflect these points. If they wish to retain their conclusion regarding a causal *zcrb1*-cilia-Wnt relationship, they will need to provide substantial additional evidence—for example, by ablating cilia formation through means other than *zcrb1* disruption and subsequently assessing Wnt pathway activation in such models, while excluding alternative mechanisms (such as cilia-independent roles of IFT proteins or direct regulation of Wnt components expression by *zcrb1*). In addition, the authors should carefully discuss any discrepancies between their findings and the existing literature.

Response: We agree with the reviewer suggestions and updated the discussion as well as Figure 7. We and the others have found that cilia and centrosome transcripts contain

minor introns that are differentially spliced in the absence of *ZCRB1*, or other minor spliceosome connected genes (Che et al., 2025). We agree with reviewer that the WNT pathway, while clearly affected by partial loss of *ZCRB1*, could be altered independently of cilia. In this respect, we have toned down our discussion and updated our proposed model. However, we do want to emphasize that WNT signaling, as well as cilia loss, can be rescued by re-introduction of the *ZCRB1* human mRNA transcript in the zebrafish, arguing that *ZCRB1* is important for both cilia formation and WNT pathway signaling. The interdependencies of these pathways is something we will continue to parse moving forward.

Additional recommendations:

It is not clear why *Wnt5a* is referred to in the text (Results, section related to Fig. 5) as a Wnt pathway inhibitor, as it is well established to be a ligand capable of activating multiple signaling pathways depending on the available receptors (PMID: 22771246).

Response: We have changed our statement to “context specific repressor *Wnt5a*” (line 315) along with the reference in the text that addresses reviewer’s comment.

In addition, the statement in the Introduction that “WNT signaling in particular can both transmit...” does not take into account the literature arguing against a direct role of cilia in Wnt pathway signal transduction. Please revise this statement accordingly.

Response: We have revised part of the manuscript based on the reviewer’s suggestion line 119-121.

Reviewer-#3:

Pirzada et al demonstrates reduction in *ZCRB1*, a component of the minor spliceosome, impacts splicing and expression of minor intron containing genes (MIGs) as would be expected. The novel aspect of this study identifies an enrichment of MIGs essential for ciliogenesis that are disrupted leading to impaired cilia formation, increased WNT signaling and developmental patterning defects. These defects can be rescued by Wnt inhibition but not cilia restoration. The use of both human cell line studies and in vivo zebrafish models support the conservation of these observations in animal development.

However I believe there are two major points that need to be addressed to strengthen the paper and support the conclusions drawn.

Major points:

Further analysis of RNAseq to support the statement that ciliogenesis MIGs are preferentially impacted by *ZCRB1* depletion.

The HEK293 RNAseq analysis which forms the basis for investigating ciliogenesis is confounded by a global effect on dysregulation of RNA metabolism and splicing components (Fig 1C & 1D). The phenotypes observed may be a consequence of the cellular stress from the reduction in RNP complexes. This limitation should be acknowledged.

Response: We would like to point the reviewer to our answers to reviewer 1 (page 1-2). We have analyzed snRNA abundance by RNA seq and Northern blots as well as levels of RNPC3 and PDCD7 by western blot analyses. It was shown earlier by Antto J Norppa et al, that these proteins interact with ZCRB1 over the snRNAs and we do not see drastic changes in their levels. It is possible that abundance of ZCRB1 drives abundance of fully assembled snRNPs but we don't see significant changes in snRNA or associated protein abundances reducing the likelihood that phenotypes are driven by stress or RNP complex reduction.

The RNAseq dataset could be further analysed to interegate all the MIGs and control for expression, intron length, splice site sequence etc. **Are cilia MIGs specifically impacted relative to other MIGs? Are cilia MIGs specifically impacted relative to other cilia (NON-MIG) genes?**

Response:

In our dataset, we identified 3,502 differential splicing events, of which 165 affect minor intron-containing genes (MIGs) with U12-type introns. Across all differential splicing (DS) event types, we identified 128 events impacting transcripts associated with primary cilia and the centrosome. Of the 116 cilia-related genes showing differential expression (DE), 29 unique genes were both alternatively spliced and expressed. Notably, more than half of these (18/29) exhibited decreased expression. These data suggest that a substantial subset of cilia/centrosome genes is subject to both transcriptional and post-transcriptional regulation, as a result of loss of ZCRB1 function. Most of these events occur in U2-type introns of MIGs. This enrichment likely reflects the interdependency of minor and major introns within the same transcript: because U12-type introns are typically rate-limiting for splicing, loss of ZCRB1 may perturb splicing of adjacent U2-type introns, resulting in broader co-transcriptional splicing defects. Overall, these results suggest that ZCRB1 deficiency has a global impact on splicing in minor intron-containing genes, beyond direct effects on U12-type introns (Table S2).

In addition the RT-PCR confirmation of MIG splicing changes should be strengthened/expanded (currently figure 2E is not convincing - see point 4 below) to a larger panel of cilia MIG genes.

Response: We have included new ciliary minor intron target *ARL16* (Fig. 2D) to strengthen our claim. We have addressed the concerns raised for *CCDC28B*, please refer to *CCDC28B* answer given to reviewer 2 (page 8).

The RNAseq should be compared to other published datasets for minor splicing depleted cell lines? Is the enrichment in impaired spicing of ciliogenesis MIGs specific to ZCRB1 depletion? Is this effect specific to HEK293 cells? Are there

datasets of cells more appropriate to studied ciliogenesis eg. RPE-1 cells?

Response: While we appreciate reviewer's suggestion it is not straightforward to compare such datasets. We have cited and referenced other publications that indicate that cilia and centrosome genes are affected by a loss of minor spliceosome genes (Che et al., 2025). The data in the referenced publication is from HeLa cells and our work indicates that loss of *ZCRB1* causes changes in cilia genes in RPE-1 cells and zebrafish as well. For this reason, we do not think that this is HEK-293 cell specific effect.

Confirmation of zebrafish development and embryonic patterning defects by ZCRB1 depletion should be extended. Currently figure 4C-D provides evidence for essentiality of ZCRB1 for zebrafish development but this is all based upon F0 CRISPR mosaic embryos generated from a single gRNA.

While the mRNA rescue experiments support the claim ZCRB1 is responsible for the developmental defects the potential for off target effects remain and independent guides or stable mutant lines are required to confirm the observation.

Response: To address the reviewer's points and validate the phenotypic specificity of our guide RNA targeting, we now show data from our expanded crisprant guide RNA injection models. First, we carried out phenotyping studies with multiple guide RNAs targeting distinct regions of the *zcrb1* transcript. 3/4 of the guide RNAs showed consistent phenotypes, which we now show in Supplemental Figure 9. Second, as shown in the original submission, we carried out injection of human WT *ZCRB1* mRNA to confirm phenotypic rescue and specificity of these phenotypes to the injected *zcrb1* guide RNA. This data remains presented in Figure 4. Finally, as recommended and requested by the reviewers, we show targeted DNA sequencing analysis and gDNA fragment analysis of the genomic region targeted by the guide RNA. These analyses show the generation of indels at the predicted guide RNA cut site, and this data is now included as Supplementary Figure 10. We hope this raises the reviewer's confidence in our approach.

Minor-points:

Should be consistent in terminology and use minor introns or U12-type introns throughout and not switch between both.

Response: We have changed our terminology for consistency in our revised manuscript.

Large spread between biological replicates in RT-qPCR data presented in Figure 1B. As these are clonal cell lines with heterozygous alleles I would not expect the variation to be so big. This may be confounded by the global dysregulation of RNA metabolism and should be commented on/addressed. **Data could be strengthened by using more than one housekeeper gene to normalise** cDNA levels. Ideal housekeeper gene candidates could be identified from RNAseq dataset.

Response: We thank the reviewer for this suggestion, we have performed the qPCR with another housekeeping gene *RPL27A* with consistent results (Fig 2S, C).

Label x-axis of figure 1D.

Response: It has been updated.

Figure 2E (&2D) - Unclear if there is a specific amplicon in WT control sample for *CCDC28B*. Not a strong example to support data. **More detail in legend or on figure would help - Label the minor intron and position of primers for the RT-PCR on the gene schematic?**

Response: We have updated the text, figures, and respective legend Fig. S6B to address these concerns. Please refer to response given to reviewer 2 for *CCDC28B* on page 8. In brief, we agree that *CCDC28B* is not the best example of intron retention, and now include additional examples for clarity.

Care should be taken to not over claim that *ZCRB1* is an essential gene for cellular fitness and survival in human cells. Data here suggests this for HEK293 cells but complete loss of *ZCRB1* was not tested in any other human cell lines. Also absence of recovering null alleles is not conclusive evidence of essentiality. **Consider altering the text to reflect this.**

Response: We agree with the reviewer that our experiments in HEK293 cells do not show enough evidence for cellular fitness and survival. We have modified the text accordingly (line 140-141 &147). However, we still want to point out that multiple independent research groups have indicated that *ZCRB1* KO is not viable (Bauer et al., 2015; Wang et al., 2015; Yilmaz et al., 2018) including the DepMap database. A recent study by Che et al., 2025 claimed to have made a *ZCRB1* KO HeLa cell line but these KO cells still retain approximately 20% protein, as can be seen in their Fig S3C. Therefore, in the introduction and discussion we make clear our claims on essentiality are based on the cells tested in other manuscripts. For our work, we have updated our text to make clear we only claim that *ZCRB1* is necessary for cellular homeostasis.

The cilia reduction in human RPE-1 cells by *ZCRB1* siRNA is compelling (figure 3A-C) but no genetic rescue was performed (siRNA resistant *ZCRB1*) to confirm phenotype was due to on target siRNA activity. **This should be performed or an alternative *ZCRB1* depletion technique such as CRISPR could be utilised as an orthogonal method to support *ZCRB1* specificity.**

Response: We confirmed the specificity of *ZCRB1*-targeting by siRNA compared to both positive and negative controls by both qPCR and western blot. We selected the siRNA used in the manuscript based on its efficacy, but all individual siRNAs tested yield the same phenotypes. The siRNA experiments were done as an addition to the HEK293 CRISPR/Cas-9 targeting of *ZCRB1*, showing consistent results on cell growth and viability.

The upregulation of WNT signaling components (figure 3E) is not quantitative, **replicates and densitometry would strengthen data. Acknowledge the phenotype could be a secondary effect of the cell stress caused by the global dysregulation of RNA metabolism seen in ZCRB1 depleted cells (see major point 1).** The most direct functional read out for WNT pathway activation would be beta catenin staining/nuclear translocation or TCF/LEF transcriptional activity. This data from human cells would complement the zebrafish studies. Rescue of the upregulated WNT signaling components in the ZCRB1 HEK293 clones (figure 3E) or reproducing the observation in a second human cell line is required to strengthen the claim that loss of ZCRB1 leads to upregulation of WNT signaling in human cells.

Response: The data have been quantified in response to reviewer's comments and can be found in Fig S8 A,B. We appreciate the further recommendations by the reviewer on dissecting the Wnt pathway. Our sequencing data indicate changes in both the canonical and non-canonical Wnt pathways in ZCRB1 heterozygous cells. Parsing the individual contributions of Wnt genes in this model is an important question, but beyond the scope of our current manuscript. We are certainly excited to address this question in forthcoming publications! In the interim, we have acknowledged the limitations of our studies and indicated where future work is needed (Lines 446-448).

Should reference new paper PMID: 41258964 (Nov. 2025) which looks at ZCRB1 disruption in liver cancer.

Response: This reference has been incorporated into our revised manuscript.

April 28, 2026

RE: Life Science Alliance Manuscript #LSA-2025-03607-TR

Dr. Sergej Djuranovic
Brown University
Molecular Biology, Cell Biology and Biochemistry
225 Dyer St Rm 642
Giuliani RNA Center
Providence, RI 02903

Dear Dr. Djuranovic,

Thank you for submitting your revised manuscript entitled "Role of the U11/U12 spliceosome gene ZCRB1 in Ciliogenesis and WNT Signaling". We apologise for the delay in communicating our decision due to editor availability issues.

Your revised manuscript was evaluated by all the original reviewers whose comments are appended below. As you will note, all the reviewers acknowledged that most of their concerns have been addressed, with Reviewers 2 and 3 commenting on the improvement in the revised manuscript.

However Reviewer 1 and 2 have some minor concerns on (1) repeat/inappropriate citations and (2) the need for moderating the conclusion or explicitly discussing limitations of the analysis in Fig. 3E and Suppl Fig 8 whilst concluding an upward trend, on phosphorylated (S1490) and total LRP6 and AXIN2, in ZCRB1-heterozygous cells. We agree that you must resolve these concerns with appropriate changes to the manuscript text.

Overall, we would be happy to publish your paper in Life Science Alliance pending resolution of all the reviewers' concerns and final revisions necessary to meet our formatting guidelines.

MANUSCRIPT ORGANIZATION AND FORMATTING:

To avoid unnecessary delays in the acceptance and publication of your paper, please read the following information carefully. Full guidelines are available on our Instructions for Authors page, <https://www.life-science-alliance.org/authors>

-We encourage you to modify the title to reflect your toned down conclusions in the revised manuscript. We offer a suggestion, "Loss of U11/U12 spliceosome gene ZCRB1 affects Ciliogenesis and WNT Signaling" or you can modify it accordingly.

-For experiments with vertebrate material, please follow our guidelines (<https://www.life-science-alliance.org/editorial-policies#animals>) to confirm that all experiments were performed in accordance with relevant guidelines and regulations, and identify the institutional and/or licensing committee approving the experiments.

-Please provide scale bars and size information for images in Figures 4B, 4C, 6A, and S11.

-Please provide a source and citation for the HEK293 Flp-In T-REx and hTERT-RPE-1 cells in the methods description (line 488).

-Please provide the primer information for qPCR evaluation of edited cells (line 508).

-Please include details for imaging of cells (excitation/emission and objective details).

-Please specify in the legend regarding the preparation of the schematic in Figure 2C as you have done for Figure 7.

-Thank you for providing a statement on data availability. Please confirm that the information provided (SRA database ID PRJNA1442218) is functional/accessible.

-Please add the X and Bluesky handles of your host institute/organization, as well as your own, and/or one of the authors, in our system.

-The titles in both the system and the manuscript file must be consistent with each other.

-Please mark the 2nd Corresponding Author in our system as well.

-Please remind the 2ndary Corresponding Author to add their Orcid ID to the system - they should have received instructions on how to do so.

-Please use the [10 author names et al.] format in your references (i.e., limit the author names to the first 10).

-Please be sure that the authorship listing and order is correct.

We welcome submissions of potential cover images for the issue of LSA in which your work would appear. If you have high quality images associated with this work, please feel free to email these, with a caption, to the journal office.

LSA encourages authors to provide a 30-60 second video where the study is briefly explained. We will use these videos on social media to promote the published paper and the presenting author (for examples, see <https://docs.google.com/document/d/1-UWCfbE4pGcDdcgzcmiuJl2XMBJnxKYeqRvLLrLSo8s/edit?usp=sharing>). Corresponding or first-authors are welcome to submit the video. Please submit only one video per manuscript. The video can be emailed to contact@life-science-alliance.org

FINAL FILES:

The following items are required for acceptance.

The license to publish form must be signed before your manuscript can be sent to production. A link to the license to publish form will be available to the corresponding author only. Please take a moment to check your funder requirements.

Thank you for your attention to these final processing requirements. Please revise and format the manuscript and upload materials as soon as you are able.

Thank you for this interesting contribution to the literature. We look forward to publishing your paper in Life Science Alliance.

Sincerely,

Sarita Hebbar, PhD
Scientific Editor
Life Science Alliance
<http://www.lsjournal.org>

Reviewer #1 (Comments to the Authors (Required)):

I have no further comments and I thank the authors for addressing all my points.

Minor points :

- I noticed few typos or inappropriate citations left (line 97, Joubert; line 99, CENATAC ; line 353, Borovina et al. 2010; line 394, Wang et al 2022; lines 480-482, ZCRB1 role/function ('s is not necessary)).
- Some references appear twice in the list.
- In the last figure (model), is there a reason why authors chose to represent multiciliated cells? In zebrafish and mammals, most of cells are "monociliated".

Reviewer #2 (Comments to the Authors (Required)):

The revised manuscript is notably improved. I would ask the authors to address the following remaining issues:

1. I find the data in Fig. 3E and Suppl. Fig. 8 unconvincing. The variability is too high to support the conclusion of 'an upregulated trend in phosphorylated (S1490) and total LRP6 and AXIN2 in our ZCRB1-heterozygous cells' (lines 261-262). While this interpretation might be cautiously considered for clone 63, the remaining clones do not appear to differ consistently from the WT control. I suggest either removing these data as inconclusive and toning down the conclusions, or explicitly discussing the limitations of the analysis, namely the inability to draw firm conclusions due to high variability.
2. The reference list contains numerous entries that appear to be duplicated (e.g., Baumgartner et al, Bernatik et al, Coschiera et al, etc.).

Reviewer #3 (Comments to the Authors (Required)):

Pirzada et al have improved their manuscript through responding to the reviewers comments. Importantly they toned down their conclusions and summary model to acknowledge that their data supports zcrb1 partial loss impacts both ciliogenesis and WNT signaling although they are unable to confirm that the impact on WNT signaling is directly downstream of the cilia phenotype.

In response to the major criticisms raised:

The analysis of the U11/U12 di-snRNP proteins and snRNA abundance in the ZCRB1 heterozygous clonal cells and the further clarification of the bioinformatic and statistical analysis performed does alleviate the concern that the changes seen in the RNAseq are not due to a global impact on RNP complexes or snRNA levels. I agree the data suggests that ZCRB1 deficiency impacts splicing in MIGs, although the impact extends beyond the minor introns and cilia related genes are enriched in the ZCRB1 impacted DEGs.

I appreciate their efforts to strengthen the zebrafish crisprant data by the use of multiple independent guides and the inclusion of gDNA fragment analysis to show evidence of gene/guide specific Indels. These new improvements along with the genetic rescue using human WT ZCRB1 mRNA provides confidence in the specificity of the phenotypes observed in their mosaic embryos.

The clarification of some data (eg. CCDC28B) and inclusions of further examples (ARL16), the refined quantification of Immunoblots and RT-qPCR are all also improvements that help strengthen the conclusions drawn.

Editor / Journal comments

-We encourage you to modify the title to reflect your toned down conclusions in the revised manuscript. We offer a suggestion, "Loss of U11/U12 spliceosome gene ZCRB1 affects Ciliogenesis and WNT Signaling" or you can modify it accordingly.

Response: The title has been modified to the toned down conclusion of our study to "Loss of U11/U12 spliceosome gene ZCRB1 leads to aberrant Ciliogenesis and WNT signalling".

-For experiments with vertebrate material, please follow our guidelines (<https://www.life-science-alliance.org/editorial-policies#animals>) to confirm that all experiments were performed in accordance with relevant guidelines and regulations, and identify the institutional and/or licensing committee approving the experiments.

Response: All of vertebrate experiments related to this study were performed in accordance to relevant guidelines and regulations.

-Please provide scale bars and size information for images in Figures 4B, 4C, 6A, and S11.

Response: Scale bars reflecting 500µm have been added to these figures.

-Please provide a source and citation for the HEK293 Flp-In T-REx and hTERT-RPE-1 cells in the methods description (line 488).

Response: This information has been updated in line 478-481.

-Please provide the primer information for qPCR evaluation of edited cells (line 508).

Response: This information can be found in relevant M&M section line 556-566.

-Please include details for imaging of cells (excitation/emission and objective details).

Response: This information has been updated in respective M&M section, line 598-601.

-Please specify in the legend regarding the preparation of the schematic in Figure 2C as you have done for Figure 7.

Response: This information has been updated in respective figure legend, line 1046-1049.

-Thank you for providing a statement on data availability. Please confirm that the information provided (SRA database ID PRJNA1442218) is functional/accessible.

Response: All of the omics data related to this study can be accessed at SRA database with bio project ID: PRJNA1442218.

-Please add the X and Bluesky handles of your host institute/organization, as well as your own, and/or one of the authors, in our system.

Response: This information has been updated.

-The titles in both the system and the manuscript file must be consistent with each other.

Response: This has been updated.

-Please mark the 2nd Corresponding Author in our system as well.

-Please remind the 2ndary Corresponding Author to add their Orcid ID to the system - they should have received instructions on how to do so.

Response: This has been updated.

-Please use the [10 author names et al.] format in your references (i.e., limit the author names to the first 10).

Response: This has been updated.

Response: This has been updated.

Reviewer #1

I have no further comments and I thank the authors for addressing all my points.

Response: We sincerely thank reviewer 1 for comments and acknowledging our efforts in addressing the concerns.

Minor points :

- I noticed few typos or inappropriate citations left (line 97, Joubert; line 99, CENATAC ; line 353, Borovina et al. 2010; line 394, Wang et al 2022; lines 480-482, ZCRB1 role/function ('s is not necessary)).

Response: These mistakes have been corrected.

- Some references appear twice in the list.

Response: These mistakes have been corrected.

- In the last figure (model), is there a reason why authors chose to represent multiciliated cells? In zebrafish and mammals, most of cells are "monociliated".

Response: We agree with the reviewer and model figure has been updated.

Reviewer #2 (Comments to the Authors (Required)):

The revised manuscript is notably improved. I would ask the authors to address the following remaining issues:

1. I find the data in Fig. 3E and Suppl. Fig. 8 unconvincing. The variability is too high to support the conclusion of 'an upregulated trend in phosphorylated (S1490) and total LRP6 and AXIN2 in our ZCRB1-heterozygous cells' (lines 261-262). While this interpretation might be cautiously considered for clone 63, the remaining clones do not appear to differ consistently from the WT control. I suggest either removing these data as inconclusive and toning down the conclusions, or explicitly discussing the limitations of the analysis, namely the inability to draw firm conclusions due to high variability.

Response: We have modified our conclusion statement to acknowledge this limitation, line 250-253 & 399-402.

2. The reference list contains numerous entries that appear to be duplicated (e.g., Baumgartner et al, Bernatik et al, Coschiera et al, etc.).

Response: These mistakes have been corrected.

Reviewer #3 (Comments to the Authors (Required)):

Pirzada et al have improved their manuscript through responding to the reviewers comments. Importantly they toned down their conclusions and summary model to acknowledge that their data supports zcrb1 partial loss impacts both ciliogenesis and WNT signaling although they are unable to confirm that the impact on WNT signaling is directly downstream of the cilia phenotype.

In response to the major criticisms raised:

The analysis of the U11/U12 di-snRNP proteins and snRNA abundance in the ZCRB1 heterozygous clonal cells and the further clarification of the bioinformatic and statistical analysis performed does alleviate the concern that the changes seen in the RNAseq are not due to a global impact on RNP complexes or snRNA levels. I agree the data suggests that ZCRB1 deficiency impacts splicing in MIGs, although the impact extends beyond the minor introns and

cilia related genes are enriched in the ZCRB1 impacted DEGs.

I appreciate their efforts to strengthen the zebrafish crisprant data by the use of multiple independent guides and the inclusion of gDNA fragment analysis to show evidence of gene/guide specific Indels. These new improvements along with the genetic rescue using human WT ZCRB1 mRNA provides confidence in the specificity of the phenotypes observed in their mosaic embryos.

The clarification of some data (eg. CCDC28B) and inclusions of further examples (ARL16), the refined quantification of Immunoblots and RT-qPCR are all also improvements that help strengthen the conclusions drawn.

Response: We sincerely thank reviewer 3 for current and previous comments and acknowledging our efforts in addressing the concerns.

May 15, 2026

RE: Life Science Alliance Manuscript #LSA-2025-03607-TRR

Dr. Sergej Djuranovic
Brown University
Molecular Biology, Cell Biology and Biochemistry
225 Dyer St Rm 642
Giuliani RNA Center
Providence, RI 02903

Dear Dr. Djuranovic,

Thank you for submitting your Research Article entitled "Loss of U11/U12 spliceosome gene ZCRB1 leads to aberrant Ciliogenesis and WNT signalling". It is a pleasure to let you know that your manuscript is now accepted for publication in Life Science Alliance. Congratulations on this interesting work.

Your article will publish open access upon publication under a CC-BY license.

DISTRIBUTION OF MATERIALS:

Again, congratulations on a very nice paper. I hope you found the review process to be constructive and are pleased with how the manuscript was handled editorially. We look forward to future exciting submissions from your lab.

Sincerely,

Sarita Hebbar, PhD
Scientific Editor
Life Science Alliance
<http://www.lsajournal.org>